# A Note on Generalization in Variational Autoencoders: How Effective Is Synthetic Data & Overparameterization?

**Tim Z. Xiao**[*]  *zhenzhong.xiao@uni-tuebingen.de*
*AI Center Tübingen, University of Tübingen, IMPRS-IS*

**Johannes Zenn**[*]  *johannes.zenn@uni-tuebingen.de*
*AI Center Tübingen, University of Tübingen, IMPRS-IS*

**Robert Bamler**  *robert.bamler@uni-tuebingen.de*
*AI Center Tübingen, University of Tübingen*

**Reviewed on OpenReview:** *https://openreview.net/forum?id=bwyHf5eery*

## Abstract

Variational autoencoders (VAEs) are deep probabilistic models that are used in scientific applications (Cerri et al., 2019; Flam-Shepherd et al., 2022; Zhou & Wei, 2020; Gondur et al., 2023) and are integral to compression (Ballé et al., 2018; Minnen et al., 2018; Cheng et al., 2020; Yang et al., 2023); yet they are susceptible to overfitting (Cremer et al., 2018). Many works try to mitigate this problem from the *probabilistic methods perspective* by new inference techniques or training procedures. In this paper, we approach the problem instead from the *deep learning perspective* by investigating the effectiveness of using synthetic data and overparameterization for improving the generalization performance. Our motivation comes from (1) the recent discussion on whether the increasing amount of publicly accessible synthetic data will improve or hurt currently trained generative models (Alemohammad et al., 2024; Bohacek & Farid, 2023; Bertrand et al., 2024; Shumailov et al., 2023); and (2) the modern deep learning insights that overparameterization improves generalization. Our investigation shows how both training on samples from a pre-trained diffusion model, and using more parameters at certain layers are able to effectively mitigate overfitting in VAEs, therefore improving their generalization, amortized inference, and robustness performance. Our study provides timely insights in the current era of synthetic data and scaling laws.

## 1 Introduction

Variational autoencoders (VAEs, (Kingma & Welling, 2014; Rezende et al., 2014)) are deep probabilistic models that consist of an encoder and a decoder. VAEs were originally proposed in the framework of generative models, where the decoder (together with a prior) models the underlying data distribution $p_{\text{data}}(\boldsymbol{x})$, while the encoder plays only an auxiliary role to speed up training. Nowadays, however, many applications use VAEs for their encoder: VAEs are widely used in science, e.g., for anomaly detection (Cerri et al., 2019) and for learning interpretable representations in quantum optics (Flam-Shepherd et al., 2022) and neuroscience (Zhou & Wei, 2020; Gondur et al., 2023). In compression, VAEs are also the dominating models due to their connection to the rate-distortion theory (Ballé et al., 2018; Minnen et al., 2018; Cheng et al., 2020; Yang et al., 2023). These applications can be compromised if the VAE overfits. More specifically, it is empirically observed that the encoder $f_\phi(\boldsymbol{x})$ is more susceptible to overfitting than the decoder (Wu et al., 2017; Cremer et al., 2018; Shu et al., 2018). One possible explanation is that a VAE is normally trained with a finite dataset, due to the ELBO objective and the training algorithm, the encoder is fed with the same data at each epoch (by contrast, the decoder is fed with random samples from the approximate posterior distribution, therefore less likely to see the same data twice during training).

---

[*]Equal contribution.

Overfitting in the encoder implies that the learned mapping $f_\phi(\boldsymbol{x})$ does not generalize well to *unseen* data, which can negatively impact the performance of generative modeling, amortized inference, and adversarial robustness. For generative modeling, as the number of training epochs increases, an overfitted VAE will have a higher evidence lower bound (ELBO) on the training set but a lower ELBO on the test set. For amortized inference, an overfitted encoder is likely to map unseen data to a suboptimal set of variational parameters. This results in a lower ELBO when compared to the ELBO obtained by directly optimizing these parameters. For robustness, an overfitted encoder often learns a less smooth $f_\phi(\boldsymbol{x})$, i.e., a small change in $\boldsymbol{x}$ can result in a large difference in the latent space. This makes VAEs vulnerable to adversarial attacks, causing realistic and hardly distinguishable inputs to yield semantically different outputs (Kuzina et al., 2022).

In learning theory, overfitting is a result of the interaction between the size of a training set and the size of a model. According to the classical understanding, given a finite training set, as the model size (and presumably also the *model complexity*) increases, its performance first increases and then decreases, resulting in a U-shape relation between test error and model size. Conversely, the modern deep learning literature shows that if we keep increasing the model size, its performance gets better again, thus exhibiting a *double descent* behavior (Belkin et al., 2019; Nakkiran et al., 2021). In this work, we investigate to what extent these findings apply to VAEs, and whether we can mitigate overfitting in VAEs and improve their generalization performance by exploiting these findings. Starting from a VAE in the overfitting regime, we investigate the effect of increasing either the size of the training set or the size of the model. Since the generalization property that we investigate is mostly affected by the *relative* size of the model compared to the dataset, **we limit the discussion in this paper to datasets that** allow us to test strongly overparameterized models (compared to the dataset size), as well as models that are underparameterized.

There is a subtlety in increasing the training data for VAEs: Since the goal of VAEs is to model $p_{\text{data}}(\boldsymbol{x})$, naive data augmentations might alter the target distribution if not carefully crafted. Hence, we ask the question: "*Can we have infinite training samples drawn from $p_{\text{data}}(\boldsymbol{x})$?*" The answer is likely to be "No", unless we have access to the true data generating process. However, there is a class of models, known as diffusion models (Sohl-Dickstein et al., 2015; Ho et al., 2020; Song et al., 2021), that can estimate $p_{\text{data}}(\boldsymbol{x})$ very well, and that can generate as many samples as we want. Contrary to diffusion models, VAEs *learn* an encoding process leading to the so-called variational information bottleneck (Alemi et al., 2016). This incentivizes VAEs to encode information into their latents efficiently by removing redundancies from the input—different from diffusion models—which makes VAEs the most important tool in compression (Ballé et al., 2018; Minnen et al., 2018; Cheng et al., 2020; Yang et al., 2023). Additionally, the learned representations are fast, controllable, and interpretable which enables semantically meaningful representations useful for applications in e.g. science (Cerri et al., 2019; Flam-Shepherd et al., 2022; Zhou & Wei, 2020; Gondur et al., 2023). We therefore investigate whether training a VAE on samples from a pre-trained diffusion model alleviates overfitting in the encoder of the VAE, thus allowing us to learn better representations. This approach can be considered as *cross-model-class distillation*, i.e., distilling from a diffusion model to a VAE.

**Contributions.** We investigate generalization performance in VAEs from a deep learning perspective by considering synthetic data and overparameterization. Particularly, we show that in the overfitting regime (1) VAEs trained on samples from pre-trained diffusion models have better generalization, amortized inference and robustness performance; (2) we do not need infinite synthetic data to gain such generalization performance; and (3) increasing the number of parameters in the layers next to the latent variable (thus increasing the latent dimension) improves the generalization performance, but increasing the number of parameters in other layers hurts performance. Our findings provide practitioners with practical guidance to avoid overfitting in VAEs. Moreover, we find indications of the double descent phenomenon in VAEs which might be explained as for non-deep models (Curth et al., 2023).

## 2 Performance Metrics for VAEs

**Variational autoencoders (VAEs).** A VAE models the data distribution $p_{\text{data}}(\boldsymbol{x})$ by assuming a generative process that first draws a latent variable $\boldsymbol{z}$ from a prior $p_{\boldsymbol{z}}(\boldsymbol{z})$ and then draws $\boldsymbol{x}$ from a conditional likelihood $p_\theta(\boldsymbol{x}\,|\,\boldsymbol{z})$, parameterized by the output of a neural network ("decoder") $g_\theta(\boldsymbol{z})$ with weights $\theta$. Given a training data distribution $p(\boldsymbol{x})$ approximating $p_{\text{data}}(\boldsymbol{x})$, naive maximum likelihood learning would

maximize the negative cross entropy $-H(p(\boldsymbol{x}), p_\theta(\boldsymbol{x})) = \mathbb{E}_{p(\boldsymbol{x})}[\log p_\theta(\boldsymbol{x})]$, where $p_\theta(\boldsymbol{x}) = \int p_\theta(\boldsymbol{x} \,|\, \boldsymbol{z}) \, p_{\boldsymbol{z}}(\boldsymbol{z}) \, d\boldsymbol{z}$. However, as the integration over $\boldsymbol{z}$ is typically intractable, VAEs resort to variational inference (Blei et al., 2017) and instead maximize the evidence lower bound (ELBO)

$$\mathcal{L}(\Theta) = \mathbb{E}_{\boldsymbol{x} \sim p(\boldsymbol{x})}\big[\mathrm{ELBO}_\Theta(\boldsymbol{x})\big] \leq -H(p(\boldsymbol{x}), p_\theta(\boldsymbol{x})), \quad \text{where} \tag{1}$$

$$\mathrm{ELBO}_\Theta(\boldsymbol{x}) = \mathbb{E}_{\boldsymbol{z} \sim q_\phi(\boldsymbol{z} \,|\, \boldsymbol{x})}\big[\log p_\theta(\boldsymbol{x} \,|\, \boldsymbol{z}) + \log p_{\boldsymbol{z}}(\boldsymbol{z}) - \log q_\phi(\boldsymbol{z} \,|\, \boldsymbol{x})\big]. \tag{2}$$

Here, the variational distribution $q_\phi(\boldsymbol{z})$ is a simple probability distribution (often a fully factorized Gaussian) whose parameters are the outputs of the inference network (or "encoder") $f_\phi(\boldsymbol{x})$, whose weights $\phi$ are learned by maximizing $\mathcal{L}(\Theta)$ jointly over all parameters $\Theta = (\theta, \phi)$.

While we would like to use $p_{\mathrm{data}}(\boldsymbol{x})$ for $p(\boldsymbol{x})$, we only have access to a finite training set $\mathcal{D}_{\mathrm{train}}$, which can lead to overfitting. We discuss three performance gaps quantifying the effects of overfitting.

**Generalization gap.**  One signal for overfitting is that a model performs better on the training set $\mathcal{D}_{\mathrm{train}}$ than on the test set $\mathcal{D}_{\mathrm{test}}$, and the test set performance decreases over training epochs. We refer to the difference between training and test set ELBO as the *generalization gap*,

$$\mathcal{G}_{\mathrm{g}} = \mathbb{E}_{\boldsymbol{x} \sim \mathcal{D}_{\mathrm{train}}}\left[\mathrm{ELBO}_\Theta(\boldsymbol{x})\right] - \mathbb{E}_{\boldsymbol{x} \sim \mathcal{D}_{\mathrm{test}}}\left[\mathrm{ELBO}_\Theta(\boldsymbol{x})\right]. \tag{3}$$

Since $\mathcal{D}_{\mathrm{train}}$ and $\mathcal{D}_{\mathrm{test}}$ both consist of samples from the same distribution $p_{\mathrm{data}}(\boldsymbol{x})$, and training maximizes the ELBO on $\mathcal{D}_{\mathrm{train}}$, the ELBO on $\mathcal{D}_{\mathrm{train}}$ is typically greater than or equal to the ELBO on $\mathcal{D}_{\mathrm{test}}$, and $\mathcal{G}_{\mathrm{g}} \geq 0$. A smaller $\mathcal{G}_{\mathrm{g}}$ corresponds to better generalization performance.

**Amortization gap.**  VAEs use amortized inference, i.e., they set the variational parameters of $q_\phi(\boldsymbol{z} \,|\, \boldsymbol{x})$ for a given $\boldsymbol{x}$ to the output of the encoder $f_\phi(\boldsymbol{x})$. At test time, we can further maximize the ELBO over the individual variational parameters for each $\boldsymbol{x}$, which is more expensive but typically results in a better variational distribution $q^*(\boldsymbol{z} \,|\, \boldsymbol{x})$. We then study the *amortization gap* (Cremer et al., 2018),

$$\mathcal{G}_{\mathrm{a}} = \mathbb{E}_{\boldsymbol{x} \sim \mathcal{D}_{\mathrm{test}}}\big[\mathrm{ELBO}_\theta^*(\boldsymbol{x})\big] - \mathbb{E}_{\boldsymbol{x} \sim \mathcal{D}_{\mathrm{test}}}\big[\mathrm{ELBO}_\Theta(\boldsymbol{x})\big] \geq 0 \tag{4}$$

where we replaced $q$ by $q^*$ in $\mathrm{ELBO}_\theta^*$. The encoder of a VAE tends to be more susceptible to overfitting than the decoder (Wu et al., 2017; Cremer et al., 2018; Shu et al., 2018). When the encoder overfits, its inference ability might not generalize to test data, which results in a lower ELBO and larger amortization gap $\mathcal{G}_{\mathrm{a}}$. A smaller $\mathcal{G}_{\mathrm{a}}$ corresponds to better generalization performance of the encoder.

**Robustness gap.**  An overfitted encoder $f_\phi(\boldsymbol{x})$ potentially learns a less smooth function such that a small change in the input space can lead to a large difference in the output space. Hence, it is easier to construct an adversarial sample $\boldsymbol{x}^{\mathrm{a}} = \boldsymbol{x}^{\mathrm{r}} + \boldsymbol{\epsilon}$ ($\|\boldsymbol{\epsilon}\|$ is within a given attack radius $\delta$) from a real data point $\boldsymbol{x}^{\mathrm{r}} \in \mathcal{D}_{\mathrm{test}}$. We construct adversarial samples by maximizing the symmetrized KL-divergence between $q_\phi(\boldsymbol{z} \,|\, \boldsymbol{x}^{\mathrm{r}})$ and $q_\phi(\boldsymbol{z} \,|\, \boldsymbol{x}^{\mathrm{a}})$ (Kuzina et al., 2022). For a successful attack, the reconstruction $\tilde{\boldsymbol{x}}^{\mathrm{a}} = g_\theta(\boldsymbol{z}^{\mathrm{a}}), \boldsymbol{z}^{\mathrm{a}} \sim q_\phi(\boldsymbol{z} \,|\, \boldsymbol{x}^{\mathrm{a}})$, is very different from the real data reconstruction $\tilde{\boldsymbol{x}}^{\mathrm{r}} = g_\theta(\boldsymbol{z}^{\mathrm{r}}), \boldsymbol{z}^{\mathrm{r}} \sim q_\phi(\boldsymbol{z} \,|\, \boldsymbol{x}^{\mathrm{r}})$, even though the inputs $\boldsymbol{x}^{\mathrm{a}}$ and $\boldsymbol{x}^{\mathrm{r}}$ are similar. Using the image similarity metric MS-SSIM (Wang et al., 2003), where higher MS-SSIM means more similar images, we define the *robustness gap* as

$$\mathcal{G}_{\mathrm{r}} = \mathbb{E}_{\boldsymbol{x}^{\mathrm{a}} \sim p(\boldsymbol{x}^{\mathrm{a}} \,|\, \boldsymbol{x}^{\mathrm{r}})} \mathbb{E}_{\boldsymbol{x}^{\mathrm{r}} \sim \mathcal{D}_{\mathrm{test}}}\big[\mathrm{MS\text{-}SSIM}\left[\boldsymbol{x}^{\mathrm{r}}, \boldsymbol{x}^{\mathrm{a}}\right] - \mathrm{MS\text{-}SSIM}\left[\tilde{\boldsymbol{x}}^{\mathrm{r}}, \tilde{\boldsymbol{x}}^{\mathrm{a}}\right]\big]. \tag{5}$$

A more robust VAE makes attacks difficult, i.e., it has a higher $\mathrm{MS\text{-}SSIM}[\tilde{\boldsymbol{x}}^{\mathrm{r}}, \tilde{\boldsymbol{x}}^{\mathrm{a}}]$ and thus a smaller $\mathcal{G}_{\mathrm{r}}$ than a less robust VAE. For more details on the construction of the attack see Appendix A.

## 3   Related Work

We group related work into generalization and overparameterization, using diffusion models, and closing the performance gaps. Work on augmentation and distillation is discussed in Section 4.

**Overparameterized models generalize.** The classical theory of machine learning suggests not to over-increase model parameters (relative to the size of the training set) to avoid fitting spurious patterns or noise in the training data, which is known as overfitting and can lead to poor generalization. However, empirical evidence in deep learning suggests that overparameterized models generalize well (Bartlett et al., 2020). Specifically, when we keep increasing the number of parameters, after the first descent, the test error will increase but a second descent will follow. This phenomenon is known as *double descent* (Belkin et al., 2019). Understanding generalization in overparameterized models is still an active research direction (Brutzkus & Globerson, 2019; Allen-Zhu et al., 2019). Double descent has been demonstrated in many deep learning models (Nakkiran et al., 2021). Our work investigates both double descent and the effect of overparameterization in the setting of VAEs by analyzing two different sets of parameters.

**Use samples from pre-trained diffusion models.** There are many recent attempts to solve various tasks with data generated by diffusion models. Azizi et al. (2023) fine-tuned a text-to-image diffusion model on ImageNet, generated state-of-the-art samples with class labels, and trained a classifier on the samples. Their result shows that the classifier trained on generated data does not outperform the classifier trained on real data. In the adversarial training setting, using generated data by diffusion models shows significant improvements on classification robustness (Croce et al., 2021; Wang et al., 2023). Tian et al. (2023) found that the visual representations learned from samples generated by text-to-image diffusion models outperform the representations learned by SimCLR (Chen et al., 2020) and CLIP (Radford et al., 2021). Alemohammad et al. (2024) trained new diffusion models with samples from previously trained diffusion models, and they found that their sample quality and diversity progressively decrease. In this work, we find that using diffusion models as data sources improves the generalization performance of VAEs.

**Improve generalization, amortized inference, and robustness in VAEs.** Cremer et al. (2018) study the amortization gap in VAEs, and they notice that overfitting in the encoder is one of the contributing factors of the gap, and that it hurts generalization. Subsequent works try to close the amortization gap by introducing new inference techniques or procedures (Marino et al., 2018; Shu et al., 2018; Zhao et al., 2019). To close the generalization gap and reduce encoder overfitting, Zhang et al. (2022) propose to freeze the decoder after a certain amount of training steps, but further train the encoder by using reconstruction samples as part of the training data. As for adversarial robustness, Kuzina et al. (2022) propose to defend a pre-trained VAE by running MCMC during inference to move $z$ towards "safer" regions in the latent space. In our case, both using synthetic data and overparameterization do not require changing the original inference procedure. They can be used orthogonally and take into account all three gaps simultaneously.

## 4 Two Paths Towards Generalization

Given a VAE that overfits and generalizes poorly, we want to know whether using more training data (**method A**); or more model parameters (**method B**) can improve its generalization performance and performance gaps (see Section 2). Here, we compare the two approaches, and discuss how using samples generated from a diffusion model is fundamentally different from naive data augmentation.

### 4.1 Method A: More Data - Diffusion Model as a $p_{\mathrm{data}}(\boldsymbol{x})$

An ideal training objective for VAEs is to maximize

$$\mathcal{L} = \mathbb{E}_{\boldsymbol{x} \sim p_{\mathrm{data}}(\boldsymbol{x})} \left[ \mathrm{ELBO}_{\Theta}(\boldsymbol{x}) \right]. \quad (6)$$

Table 1: Training distributions for VAEs (see Figure 1), and whether we (1) can sample arbitrarily many unique samples from (2) an accurate approximation of $p_{\mathrm{data}}(\boldsymbol{x})$.

However, we only have $\mathcal{D}_{\mathrm{train}}$ as a finite approximation of $p_{\mathrm{data}}(\boldsymbol{x})$. Hence, we normally maximize

$$\mathcal{L} = \mathbb{E}_{\boldsymbol{x} \sim \mathcal{D}_{\mathrm{train}}} \left[ \mathrm{ELBO}_{\Theta}(\boldsymbol{x}) \right], \quad (7)$$

| approx. by | $\mathcal{D}_{\mathrm{train}}$ | $p_{\mathrm{aug}}(\boldsymbol{x}')$ | $p_{\mathrm{DM}}(\boldsymbol{x}')$ |
|---|---|---|---|
| (1) $\infty$ unique samples | ✗ | ✓ | ✓ |
| (2) accurate | ✓ | ✗ | ✓ |

which can lead to overfitting. Rather than focusing on model architectures or training techniques as in prior works (Shu et al., 2018; Zhao et al., 2019; Zhang et al., 2022), we aim to mitigate overfitting by seeking a better approximation for $p_{\mathrm{data}}(\boldsymbol{x})$ than $\mathcal{D}_{\mathrm{train}}$. An ideal training data distribution should enable us to

(1) sample an unlimited amount of **unique samples** to avoid overfitting; and it should be

(2) **an accurate approximation of** $p_{\text{data}}(\boldsymbol{x})$, i.e., we are indeed modeling $p_{\text{data}}(\boldsymbol{x})$ rather than some different distribution (in practice, it needs to be an accurate model of $\mathcal{D}_{\text{train}}$).

Our hypothesis is that a good diffusion model[1] that has been pre-trained on $\mathcal{D}_{\text{train}}$ satisfies these two criteria: (1) we can generate unlimited samples from it, and (2) its training objective is designed to model $p_{\text{data}}(\boldsymbol{x})$. Therefore, we investigate training VAEs using a pre-trained diffusion model $p_{\text{DM}}(\boldsymbol{x}')$ instead of $\mathcal{D}_{\text{train}}$ as an approximation of the underlying distribution $p_{\text{data}}(\boldsymbol{x})$, i.e., by maximizing

$$\mathcal{L} = \mathbb{E}_{\boldsymbol{x}' \sim p_{\text{DM}}(\boldsymbol{x}')} \left[ \text{ELBO}_{\Theta}(\boldsymbol{x}') \right]. \tag{8}$$

We denote this method DMaaPx, short for "Diffusion Model as a $p_{\text{data}}(\boldsymbol{x})$". Figure 1 illustrates the intuition behind DMaaPx. The blue dots represent the finite data set $\mathcal{D}_{\text{train}}$. They are i.i.d. samples from the unknown underlying data distribution $p_{\text{data}}(\boldsymbol{x})$ which is shown by the dark-edged region. The green regions represent $p_{\text{DM}}(\boldsymbol{x}')$. We use shaded areas (green), not dots, to highlight that $p_{\text{DM}}(\boldsymbol{x}')$ models a continuous distribution which we can generate infinitely many samples from.

Diffusion models for data types other than images are less explored and might not accurately approximate $p_{\text{data}}(\boldsymbol{x})$ and therefore might not satisfy criterion (2). Moreover, due to the data processing inequality, information on $p_{\text{data}}(\boldsymbol{x})$ captured by a diffusion model trained on $\mathcal{D}_{\text{train}}$ cannot exceed the information contained in $\mathcal{D}_{\text{train}}$ (but for injected information via inductive biases). In reality, state-of-the-art diffusion models cannot fit $\mathcal{D}_{\text{train}}$ perfectly. Many recent works observe in both image and text settings that training generative models on generated data degrades the overall performance (Alemohammad et al., 2024; Bohacek & Farid, 2023; Bertrand et al., 2024; Shumailov et al., 2023). Hence, the continuity we gain by replacing $\mathcal{D}_{\text{train}}$ with $p_{\text{DM}}(\boldsymbol{x}')$ comes at the cost of potentially losing some amount of information contained in $\mathcal{D}_{\text{train}}$.

### 4.1.1 Difference Between Data Augmentation and DMaaPx

Data augmentation[2] has a similar goal as DMaaPx as both approaches aim to increase amount and diversity of training data. Training a VAE via data augmentation amounts to maximizing

$$\mathcal{L} = \mathbb{E}_{\boldsymbol{x} \sim \mathcal{D}_{\text{train}}} \left[ \mathbb{E}_{p_{\text{aug}}(\boldsymbol{x}' \,|\, \boldsymbol{x})} \left[ \text{ELBO}_{\Theta}(\boldsymbol{x}') \right] \right], \tag{9}$$

where $p_{\text{aug}}(\boldsymbol{x}' \,|\, \boldsymbol{x})$ is a distribution over some hand-crafted transformations (e.g., scalings and rotations) of $\boldsymbol{x} \sim \mathcal{D}_{\text{train}}$.

Like DMaaPx, data augmentation replaces $\mathcal{D}_{\text{train}}$ with a continuous distribution $p_{\text{aug}}(\boldsymbol{x}') = \mathbb{E}_{\boldsymbol{x} \sim \mathcal{D}_{\text{train}}}[p_{\text{aug}}(\boldsymbol{x}' \,|\, \boldsymbol{x})]$. But $p_{\text{aug}}(\boldsymbol{x}')$ can be a less accurate approximation of $p_{\text{data}}(\boldsymbol{x})$ than $p_{\text{DM}}(\boldsymbol{x})$ (Table 1). Firstly, typical data augmentation techniques generate new training points $\boldsymbol{x}'$ by conditioning on a *single* original data point $\boldsymbol{x}$ (i.e., the nested expectations in Eq. (9)). Thus, $p_{\text{aug}}(\boldsymbol{x}')$ cannot interpolate between points in $\mathcal{D}_{\text{train}}$ (i.e., the gaps between purple regions in Figure 1). By contrast, in DMaaPx, each training data point $\boldsymbol{x}' \sim p_{\text{DM}}(\boldsymbol{x}')$ is drawn from a diffusion model trained on the entire dataset $\mathcal{D}_{\text{train}}$. Hence, each $\boldsymbol{x}'$ is effectively conditioned on the full $\mathcal{D}_{\text{train}}$.

Secondly, the transformations used for $p_{\text{aug}}(\boldsymbol{x}' \,|\, \boldsymbol{x})$ are drawn from a manually curated catalog. This catalog is heavily based on prior assumptions regarding *invariances* in the data type under consideration, which can introduce bias. In

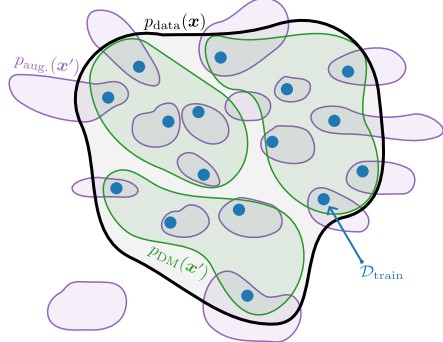

Figure 1: Training distributions. $p_{\text{aug}}(\boldsymbol{x}') = \mathbb{E}_{\boldsymbol{x} \sim \mathcal{D}_{\text{train}}}[p_{\text{aug}}(\boldsymbol{x}' \,|\, \boldsymbol{x})]$ only extrapolates from individual data points $\boldsymbol{x} \sim \mathcal{D}_{\text{train}}$ and has density outside the support of $p_{\text{data}}(\boldsymbol{x})$ (e.g., when flipping the digit "2"). By contrast, $p_{\text{DM}}(\boldsymbol{x}')$ can interpolate between data $\boldsymbol{x} \sim \mathcal{D}_{\text{train}}$.

---

[1]An *unconditional* diffusion model, as opposed to a conditional one like Stable Diffusion (Rombach et al., 2022).

[2]Using samples from generative models for training is sometimes considered as a kind of data augmentation in the context of supervised learning (Yang et al., 2022). In the present work, we deliberately separate generative models from the broader sense of data augmentation, and we consider data augmentation in a narrow sense such that the augmented data is an output of some transformation conditioned on an single $\boldsymbol{x} \in \mathcal{D}_{\text{train}}$.

practice, one has to make assumptions and decide whether the (unknown) true data distribution $p_{\text{data}}(\boldsymbol{x})$ is invariant under the considered transformations. For instance, with images, we assume invariance to minor translations, hue shifts, and zooms. For data types other than images, such as neural activations or molecules, manually crafting the appropriate set of augmentations might be less obvious. This manual design of augmentations may result in problems of (i) not modeling the *full* extend of $p_{\text{data}}(\boldsymbol{x})$ or (ii) modeling density *outside* of $p_{\text{data}}(\boldsymbol{x})$. Figure 1 depicts both: problem (i) corresponds to "empty" space between areas of $p_{\text{aug}}(\boldsymbol{x}')$ (purple); problem (ii) corresponds to density of $p_{\text{aug}}(\boldsymbol{x}')$ outside of $p_{\text{data}}(\boldsymbol{x})$. DMaaPx eliminates these assumptions, which makes the method more resilient against human bias (but less interpretable).

In summary, while traditional data augmentation introduces diversity based on *invariances* in the data, DMaaPx uses an expressive generative model to extrapolate from the *empirical* diversity of the data.

### 4.1.2 A Distillation Perspective

Using samples from a pre-trained diffusion model to train a VAE can also be viewed from a distillation perspective (Hinton et al., 2015). Distillation describes the process of transferring knowledge from a large model to a smaller one. In practice, distillation is often used because a smaller model is *less expensive* to be deployed in production. On the contrary, here we consider transferring knowledge between models designed with *different modeling assumptions or structures*. We refer to this as *cross-model-class distillation*, and the conventional usage of distillation as *within-model-class distillation*. While both types seek to transfer knowledge from a source to a target model, cross-model-class distillation emphasizes more on enhancing functionalities that are unique to the target model rather than mirroring the capabilities shared with the source model. For instance, Frosst & Hinton (2017) distill a neural network into a soft decision tree to improve its unique capability of providing explanations to decisions.

DMaaPx belongs to cross-model-class distillation, i.e., it distills diffusion models to VAEs. The goal of DMaaPx is not to rival diffusion models in sample quality, but rather to improve the unique capabilities of VAEs such as its variational bottleneck (Alemi et al., 2016) for learning fast, controllable, and interpretable representations (see Section 1 for a discussion). Hence, DMaaPx fundamentally differs from approaches that train VAEs on samples produced by VAEs (Shumailov et al., 2023), or diffusion models on outputs of diffusion models (Alemohammad et al., 2024), which belongs to within-model-class distillation.

## 4.2 Method B: More Parameters

The success of deep learning and empirical evidence (Nakkiran et al., 2021) show that increasing the number of parameters in a model can improve its generalization performance. However, increasing the number of parameters in one part of the model can have a different effect than increasing the number of parameters in another part of the model. Different sets of parameters form distinct *complexity axes*, along which generalization performance can behave in different ways.

In practice, increasing the number of parameters in VAEs is usually done by changing two hyperparameters, which control two distinct sets of model parameters, $\Theta = \Theta_{\boldsymbol{z}} \cup \Theta_{\neg \boldsymbol{z}}$. Here, $\Theta_{\boldsymbol{z}}$ contains the parameters that determine the latent dimension $\mathrm{d}_{\boldsymbol{z}}$ and interact directly with $\boldsymbol{z}$, i.e., the weights in the last layer of the encoder $f_\phi(\boldsymbol{x})$ and the first layer of the decoder $g_\theta(\boldsymbol{z})$. Changing the number $|\Theta_{\boldsymbol{z}}|$ of parameters in $\Theta_{\boldsymbol{z}}$ changes the dimension $\mathrm{d}_{\boldsymbol{z}}$ of $\boldsymbol{z}$. We denote the remaining parameters as $\Theta_{\neg \boldsymbol{z}}$.

We investigate the effect of increasing $|\Theta_{\boldsymbol{z}}|$ and $|\Theta_{\neg \boldsymbol{z}}|$ separately as the increase has different implications for each parameter group. For both sets, we only consider changing the width of the VAE, not the depth. For a VAE with mean field assumption, for example, increasing $|\Theta_{\boldsymbol{z}}|$ implies that we update our belief about the underlying data generative process, which we now believe to involve more independent factors (hence, we increase $\mathrm{d}_{\boldsymbol{z}}$ to model these additional independent factors).

**Explaining Double Descent with Multiple Complexity Axes.** With the two complexity axes $|\Theta_{\boldsymbol{z}}|$ and $|\Theta_{\neg \boldsymbol{z}}|$ mentioned above, we might potentially observe the phenomenon of double descent in VAEs. Recent work by Curth et al. (2023) tries to explain double descent (see Section 1) in non-deep models such as trees and linear regression. They suggest that double descent is the result of plotting the test error along

an aggregated model complexity axis (i.e., the total number of parameters), even though the raw parameters are increased first along one complexity axis and then along another. In other words, the test error along each axis shows the classical U-shape, but plotting them into the same model complexity axis leads to the shape of double descent. We refer readers to (Curth et al., 2023, Figure 1) for an illustration. Whether this explanation applies to *deep* double descent is still unknown. Nevertheless, it highlights that increasing different sets of model parameters can impact the model differently.

## 5    Experiments and Discussions

In this section, we introduce the experimental setup and evaluate the generalization performance as well as the performance gaps (see Section 2) for both using more training data (**method A**) and more parameters (**method B**). We then evaluate and discuss the effects of these two methods.

### 5.1    Experimental Setup

**Datasets.**    We evaluate both methods A & B, on CIFAR-10 (Krizhevsky et al., 2009). We further evaluate method A on BinaryMNIST (LeCun et al., 1998) and FashionMNIST (Xiao et al., 2017), which show consistent results with CIFAR-10 (see Appendix B.3). Due to computation constraints, we did not evaluate method B on these two easier datasets. As a preparation for DMaaPx in method A, we train a diffusion model $p_{\text{DM}}(\boldsymbol{x}')$ on each training set $\mathcal{D}_{\text{train}}$, which we then use to generate the training data for the VAEs. We use the implementation by Karras et al. (2022). Further details can be found in Appendix D.

**VAE architectures.**    We assume fixed Gaussian priors $p_{\boldsymbol{z}}(\boldsymbol{z}) = \mathcal{N}(\boldsymbol{0}, \mathbf{I})$. For the conditional likelihood $p_{\theta}(\boldsymbol{x} \,|\, \boldsymbol{z})$, we use a discretized mixture of logistics (MoL; (Salimans et al., 2017)). For the inference model $q_{\phi}(\boldsymbol{z} \,|\, \boldsymbol{x})$, we use fully factorized Gaussian distributions whose means and variances are the outputs of the encoder. Both the encoder and the decoder consist of convolutional layers with residual connections. The number of output channels for the encoder and the number of input channels for the decoder (i.e., the latent dimension of $\boldsymbol{z}$) is denoted as $\mathrm{d}_{\boldsymbol{z}} = m_z \times 64$, where $m_z$ is an integer multiplier. The number of the internal channels, which governs the rest of the parameters, is denoted as $n_c$. Hence, $|\Theta_{\boldsymbol{z}}| = \mathrm{d}_{\boldsymbol{z}} \times c_1$ and $|\Theta_{\neg\boldsymbol{z}}| = n_c \times c_2$, where $c_1$ and $c_2$ are the appropriate constants. For more details on the network architectures, hyperparameters, and training please consult Appendix B.1.

**Baselines.**    The default models for method A and the baseline for method B use $m_z = 1$ and $n_c = 256$. For method A, we compare VAEs trained with DMaaPx against three other models trained on: (i) repetitions of $\mathcal{D}_{\text{train}}$ ("Normal Training"); (ii) carefully tuned augmentation for $\mathcal{D}_{\text{train}}$ ("Aug.Tuned"); and (iii) plausible augmentation for images in general ("Aug.Naive"). The results are averaged over 3 random seeds. Note that "Aug.Naive" is not tuned towards a given training dataset $\mathcal{D}_{\text{train}}$ and can result in out-of-distribution data, e.g. a horizontally flipped digit "2" for MNIST. This mimics situations where the invariances of the data modality are less clear than for images. We report the specific augmentations used for Aug.Tuned and Aug.Naive in Appendix E. For method B, we compare VAEs trained with various $m_z \in \{1, \ldots, 64\}$ and $n_c \in \{1, \ldots, 512\}$. When documenting the training progress, the term "epoch" usually refers to one complete pass over $\mathcal{D}_{\text{train}}$. For DMaaPx, this term is not applicable since it can sample unlimited data from $p_{\text{DM}}(\boldsymbol{x}')$. Therefore, we measure training progress of DMaaPx in "effective epochs", which count multiples of $|\mathcal{D}_{\text{train}}|$ training samples. We train all models for 1000 (effective) epochs.

### 5.2    Training with synthetic data improves generalization and minimizes the gaps

Figure 2 shows the performance for generalization (left), amortized inference (mid), and robustness (right) across normal training, augmentations, and DMaaPx. We observe that DMaaPx has the highest ELBO on $\mathcal{D}_{\text{test}}$ (left, solid green) and the smallest generalization, amortization, and robustness gaps (Eqs. 3-5). Augmentation provides competitive improvements, but is not as good as DMaaPx. This implies that VAEs trained with DMaaPx approximate the underlying distribution $p_{\text{data}}(\boldsymbol{x})$ better than those trained on $\mathcal{D}_{\text{train}}$ solely, or on augmented data, as suspected in Section 4.1.1.

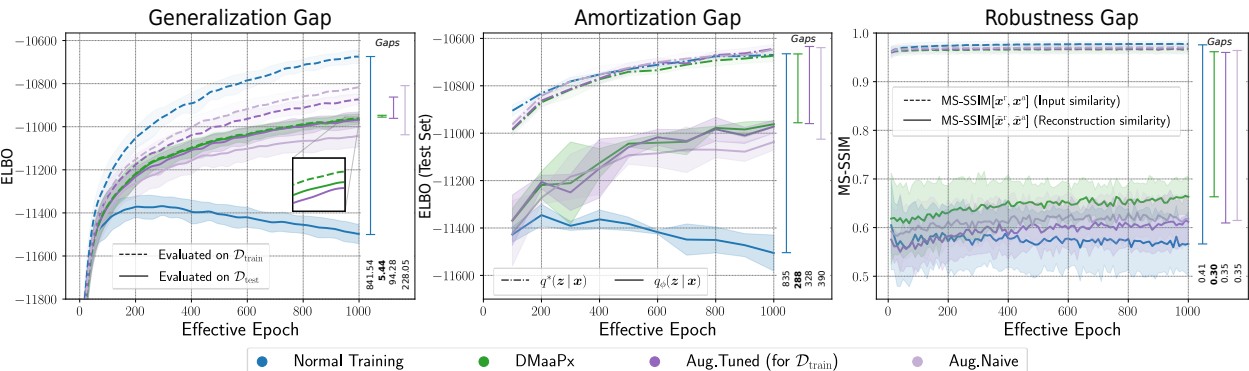

Figure 2: Generalization (left), amortized inference (mid), and robustness (right) performance for VAEs trained with Eqs. (7)-(9). Being slightly better than augmentations, DMaaPx consistently has the best performance on the test set and the smallest generalization, amortization, and robustness gap.

We also evaluate whether the improvements in generalization from DMaaPx and augmentation propagate to downstream tasks for which VAEs are better suited than diffusion models. We assess the representation learning performance by classifying the mean $\boldsymbol{\mu}$ of $q_\phi(\boldsymbol{z} \mid \boldsymbol{x})$ for each $\boldsymbol{x}$. We also test both the quality and robustness of the learned representations by using CIFAR-10-C (Hendrycks & Dietterich, 2019), a dataset containing 19 versions of the CIFAR-10 test set, each corrupted with a different corruption. For every test set, we encode each image to its $\boldsymbol{\mu}$ and then split all representations into two separate subsets. We use one subset to train a classifier and test on the other. Our experiments include four classifiers: logistic regression (LR), a support vector machine (Boser et al., 1992) with linear kernel (SVM-L), an SVM with radial basis function kernel (SVM-RBF), and $k$-nearest neighbors (kNN) with $k = 5$. Figure 4 (left) shows that classifiers trained with representations from DMaaPx (green) outperform normal training on average (details in Appendix G). By contrast, augmentation (purple) hurts performance.

**Comparison to Prior Work on Modifying the Training Procedures.** In Figure 5 we compare the generalization performance of DMaaPx to the previously proposed Reverse Half Asleep (RHA) method of (Zhang et al., 2022). Unlike DMaaPx, which focuses on the training data, RHA modifies the training procedure of a VAE such that first a VAE is trained 500 epochs, and then the decoder is fixed on only the encoder is trained on samples of the decoder for an additional 500 epochs. We find that DMaaPx, being conceptually much simpler, outperforms RHA significantly.

## 5.3 Smoothness of Latent Space

We hypothesize that an overfitted encoder often learns a less smooth $f_\phi(\boldsymbol{x})$ meaning that a small change in $\boldsymbol{x}$ can result in a large difference in the latent space. Here, we report an exemplary density for an interpolation in the latent space of a VAE trained on CIFAR-10 with normal training and DMaaPx. We map two data samples $\boldsymbol{x}_0$ and $\boldsymbol{x}_1$ from the test set of CIFAR-10 to their latent means using the encoder $\boldsymbol{z}_0 = \boldsymbol{\mu}_0 = f_\phi^{\boldsymbol{\mu}}(\boldsymbol{x}_0)$ where $f_\phi^{\boldsymbol{\mu}}(\boldsymbol{x})$ denotes the encoder returning only the mean of $q_\phi(\boldsymbol{x})$. We interpolate linearly in the latent space and evaluate $\log q_\phi(\boldsymbol{z}_\alpha \mid \tilde{\boldsymbol{x}}_\alpha)$ where $\boldsymbol{z}_\alpha = \alpha \boldsymbol{z}_0 + (1-\alpha)\boldsymbol{z}_1, \alpha \in [0,1]$ and $\tilde{\boldsymbol{x}}_\alpha$ is a reconstruction of $\boldsymbol{z}_\alpha$.

We find that, firstly, the density is higher for DMaaPx compare to normal training and, secondly, the latent space appears to be smoother for DMaaPx compared to normal training.

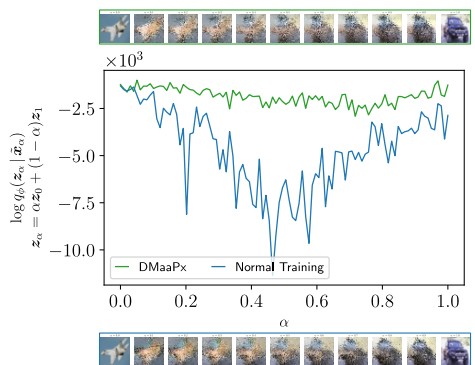

Figure 3: Density of $\log q_\phi$ evaluated on a line that linearly interpolates between two data samples from the test set of CIFAR-10. DMaaPx is smoother than normal training.

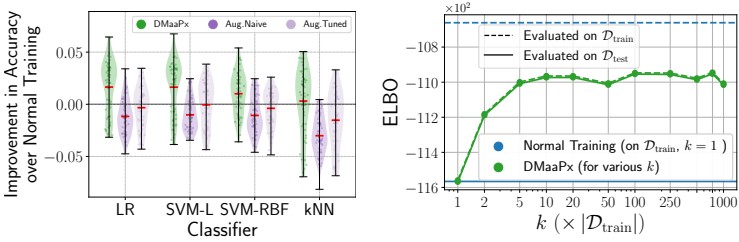
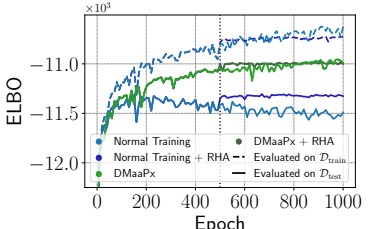

Figure 4: Left: Improvements in classification accuracy over normal training for various classifiers trained on the latent representations of CIFAR-10-C. Each column contains $19 \times 3$ points (i.e., 19 corruptions, each VAE trained with 3 random seeds). Right: Generalization performance as a function of the amount $k$ of training data sampled from a diffusion model. Horizontal blue lines show baseline performance (VAE trained directly on $\mathcal{D}_{\text{train}}$). All VAEs were trained for 1000 effective epochs. $k \approx 10$ suffices.

Figure 5: Comparison between DMaaPx and Reverse Half Asleep (RHA) (Zhang et al., 2022). Dotted vertical line shows epoch when decoder is frozen for RHA. "Normal Training + RHA" improves upon baseline but it is still worse than DMaaPx. Also, adding RHA on top of DMaaPx does not lead to improvements.

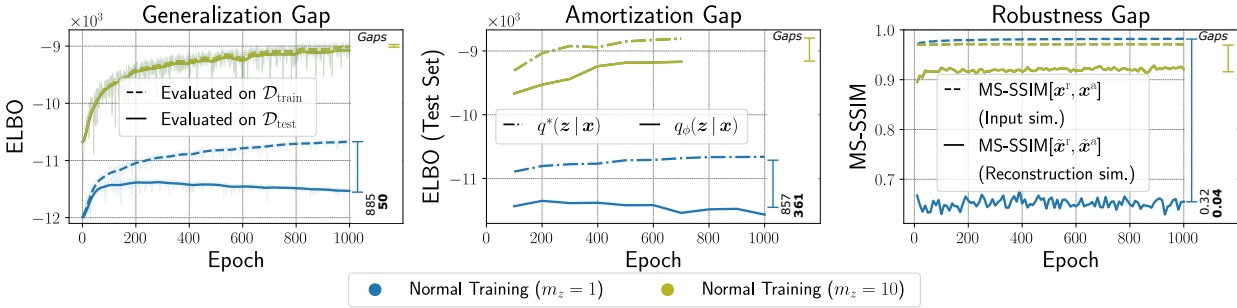

Figure 6: Generalization (left), amortized inference (mid), and robustness (right) performance for models with different $m_z$ but the same $n_c = 256$, using normal training. Due to computation constraints, we only plot the amortization gap for $m_z = 10$ up to epoch 700.

## 5.4 Do we need "unlimited" synthetic data?

With the diffusion model in DMaaPx, we can train a VAE on unlimited samples from $p_{\text{DM}}(\boldsymbol{x}')$, which improves various performance metrics as demonstrated above. However, generating lots of samples from a diffusion model is computationally expensive (see also Appendix D.2). Therefore, we further explore: "*Do we really need an infinite number of samples?*" The answer, reassuringly, is "*No*".

Figure 4 (right) shows the generalization performance of DMaaPx on CIFAR-10 where the amount of training data is restricted to $k \times |\mathcal{D}_{\text{train}}|$ with $k = 1, \ldots, 1000$. After $k$ effective training epochs samples repeat. All models are trained for 1000 effective epochs. The horizontal blue lines represent the generalization gap of normal VAE training (on $\mathcal{D}_{\text{train}}$ and $k = 1$) at epoch 1000 from Figure 2 (left). For $k = 1$, DMaaPx matches normal training on CIFAR-10. The ELBO plateaus for $k \geq 10$, indicating that approximately $10 \times |\mathcal{D}_{\text{train}}|$ samples offer a similar generalization as $1000 \times |\mathcal{D}_{\text{train}}|$ samples.

## 5.5 Increasing $|\Theta_{\boldsymbol{z}}|$ improves generalization and minimizes the gaps

Figure 6 shows the generalization (left), amortized inference (mid), and robustness (right) performance between the baseline parameter setting ($m_z = 1$, $n_c = 256$) and the setting with an increased amount of parameters ($m_z = 10$, $n_c = 256$). Note that we only increase $|\Theta_{\boldsymbol{z}}|$ here and keep $|\Theta_{\neg \boldsymbol{z}}|$ unchanged. We observe that $m_z = 10$ has the higher ELBO on $\mathcal{D}_{\text{test}}$ (left), and the smaller generalization, amortization, and

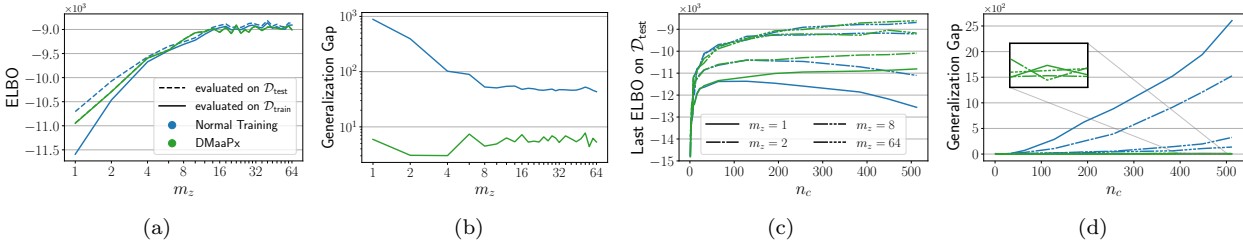

Figure 7: (a) and (b) show the changes of ELBOs and generalization gaps while fixing $n_c = 256$ and increasing $m_z$. (c) and (d) show the effects of increasing $n_c$ while keeping $m_z$ fixed. Besides normal training, we also evaluate DMaaPx in this setting.

robustness gaps (Eqs. 3-5). The resulting performance improvements are similar to DMaaPx in Figure 2, where we also see a higher ELBO test and smaller gaps.

We also investigate whether continuing to increase $m_z$ will lead to better or worse generalization performance. Figure 7 (a, b) show trends of the ELBOs on $\mathcal{D}_{\text{test}}$ and the generalization gaps (Eq. (3)) as the result of increasing $m_z$ from 1 to 64. We see that increasing $m_z$ leads to higher test ELBO for normal training until $m_z = 16$, after which the test ELBO stays around the same level. We also find that the benefit of using synthetic data (i.e., DMaaPx) for improving test ELBO diminishes as $m_z$ increases. In particular, after $m_z = 16$, both normal training and DMaaPx have similar test ELBOs. However, we want to highlight that DMaaPx has always a smaller generalization gap than normal training, even after $m_z = 16$. Small generalization gaps are desirable as they imply that the ELBO evaluated on the training set can be used to predict the performance of a VAE on a held-out test set.

## 5.6   Increasing $|\Theta_{\neg z}|$ might hurt generalization

We further investigate the effect of increasing parameters along the other complexity axis (i.e., $|\Theta_{\neg z}|$). Figure 7 (c, d) show the test ELBOs and the generalization gaps (Eq. (3)) as functions of $n_c = 1$ to 512. Subplot (c) shows that when $m_z = 1$ and $m_z = 2$, increasing $n_c$ results in the classical U-shape of overfitting for normal training, where the test performance first grows then drops. Increasing $m_z$ further to 8 and 64 tilts the right-hand side of the U-shaped curve upward and eliminates the symptom of declining generalization w.r.t. larger $n_c$, which aligns with Figure 7 (a). In particular, when $m_z = 64$, we see improved on generalization as we keep increasing $n_c$, even though the improvement from enlarging $m_z$ is already saturated (as showed in Figure 7 (a)). This implies that: (1) the two sets of parameters $\Theta_z$ and $\Theta_{\neg z}$ do indeed have different meanings; (2) increasing $|\Theta_{\neg z}|$ when $|\Theta_z|$ is small hurts generalization performance; (3) increasing both $|\Theta_{\neg z}|$ and $|\Theta_z|$ leads to better generalization than increasing only one of them.

The plot for generalization gaps (Figure 7 (d)) shows that, by increasing $n_c$, the gaps for normal training become larger. However, increasing $m_z$ reduces the gaps (which aligns with Figure 7 (b)). In addition, both plots (Figure 7 (c) and (d)) show that using synthetic data (i.e., DMaaPx) on top of adding parameters does not hurt and often improves generalization. Especially, using DMaaPx constantly results in near-zero generalization gaps (see the overlapping flat lines in Figure 7 (d)).

## 5.7   Double descent in VAEs?

Section 5.5 and Section 5.6 indicate that the modern wisdom of "larger models generalize better" does indeed apply to VAEs. Therefore, we are curious whether the double descent phenomenon also applies here. Figure 7 (a) and (c) show that if we fix one complexity axis and only increase parameters along the other complexity axis, the test ELBO exhibits either a U-shaped or a L-shaped curve, which does not look like double descent. However, one can artificially construct a double descent behavior if one sequentially increases parameters first along one axis and then along the other axis, and plots the test ELBO against the overall model complexity (i.e., the total number of parameters). This is illustrated in Figure 8. The figure shows the negative test ELBO, which shows three phases:

(1) we start with a fixed $m_z = 1$ and increase $n_z$ from 1 to 256 (the cyan U-curve); (2) we then fixed $n_z = 256$ and start increasing $m_z$ from 1 to 64 (the brown vertical line); (3) lastly, we fixed $m_z = 64$ and increase $n_z$ from 256 to 512 (the cyan horizontal line). Phase (2) seems vertical due to the fact that increasing $m_z$ by 1 adds much less raw parameters than increasing $n_z$ by 1.

Now we can see indications of a double descent behavior as the negative test ELBO goes down, up, and down again as we keep increasing the total number of parameters in the model. While the double-descent behavior was enforced artificially here by the choice of trajectory in model complexity space, it is conceivable that a trained models may allocate resources in a similar way while following a more natural trajectory. This aligns with our discussion in Section 4.2 and the work by Curth et al. (2023), who point out that double descent in non-deep models is due to increasing parameters along distinct complexity axis sequentially but plotting them on a combined complexity axis.

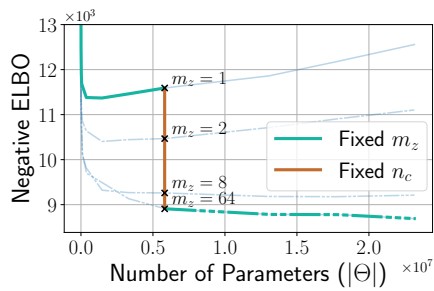

Figure 8: By increasing the total number of parameters first along the complexity axis $|\Theta_{\neg z}|$ (i.e., larger $n_c$, cyan), then along the complexity axis $|\Theta_z|$ (i.e., larger $m_z$, brown line), then back to $|\Theta_{\neg z}|$ (cyan), we see indications of double descent.

## 6    Conclusion

In this paper, we study overfitting and generalization in VAEs by investigating the effects of (a) training with synthetic data and (b) increasing the number of model parameters, both of which are timely questions to be investigated in generative models. We find that training on samples from a diffusion model that was pre-trained on the training dataset consistently improves performance. Generally, increasing the amount of parameters also helps. However, when constrained by resources, one should prioritize the parameters related to the latent dimension over the parameters that do not directly impact the latent dimension. Applying (a) and (b) together can further improve the performance. Additionally, we find indications of a double-descent phenomenon in VAEs which we defer to future work for an investigation in detail.

**Acknowledgments**

The authors would like to thank Anna Kuzina, Zhen Liu, and Weiyang Liu for helpful discussions. Funded by the Deutsche Forschungsgemeinschaft (DFG, German Research Foundation) under Germany's Excellence Strategy – EXC number 2064/1 – Project number 390727645. This work was supported by the German Federal Ministry of Education and Research (BMBF): Tübingen AI Center, FKZ: 01IS18039A. Robert Bamler acknowledges funding by the German Research Foundation (DFG) for project 448588364 of the Emmy Noether Programme. The authors thank the International Max Planck Research School for Intelligent Systems (IMPRS-IS) for supporting Tim Z. Xiao and Johannes Zenn.

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

# Appendix

## Table of Contents

## A   Details on The Adversarial Attack

We follow Kuzina et al. (Kuzina et al., 2022) and construct an unsupervised encoder attack that optimizes the pertubation $\boldsymbol{\epsilon}$ to incur the largest possible change in $q_\phi(\cdot \mid \boldsymbol{x})$,

$$\boldsymbol{\epsilon} = \underset{\|\boldsymbol{\epsilon}\|_\infty \leq \delta}{\arg\max} \ \mathrm{SKL}\left[q_\phi(\cdot \mid \boldsymbol{x}^{\mathrm{r}} + \boldsymbol{\epsilon}) \,\|\, q_\phi(\cdot \mid \boldsymbol{x}^{\mathrm{r}})\right] \tag{10}$$

where SKL denotes the symmetric Kullback-Leibler divergence (Kullback & Leibler, 1951). We optimize $\boldsymbol{\epsilon}$ for $n^\epsilon$ iterations with projected gradient descent utilizing a learning rate of $\eta$. The robustness gap (see Section 2) is computed over $n^{\mathrm{r}}$ real images and $n^{\mathrm{a}}$ random seeds. The exact hyperparameters can be found in Table 2.

Table 2:  Hyperparameters for the unsupervised encoder attack.

|               | BinaryMNIST | FashionMNIST | CIFAR-10 |
|---------------|-------------|--------------|----------|
| $n^{\mathrm{r}}$ | 50       | 50           | 20       |
| $n^{\mathrm{a}}$ | 10       | 10           | 10       |
| $n^\epsilon$  | 50          | 50           | 100      |
| $\eta$        | 1.0         | 1.0          | 1.0      |
| $\delta$      | 0.1         | 0.1          | 0.05     |

## B   Further Details on VAEs and Ablations

In this section we present further details on the VAE models and ablation studies.

### B.1   Architectures and Training Cost

This section provides a detailed description of the VAE models utilized throughout the paper.

Table 3:  Details on VAE architectures ordered by dataset. MoL refers to the discretized mixture of logistics likelihood model (Salimans et al., 2017).

| dataset      | likelihood              | architecture     |
|--------------|-------------------------|------------------|
| BinaryMNIST  | Bernoulli               | fully-connected  |
| FashionMNIST | fixed-variance Gaussian | fully-connected  |
| FashionMNIST | MoL                     | fully-connected  |
| CIFAR-10     | MoL                     | residual network |

We consider a fully-connected architecture and a residual architecture (He et al., 2016). Table 3 gives more details on the likelihood model and architecture. For all VAEs, we choose the hyperparameters of the VAE models by consulting existing literature. The MoL likelihood for FashionMNIST is discussed in Appendix B.

The fully-connected architecture maps from an input dimension of $32^2$ to a hidden dimension of 512. After a hidden layer mapping from 512 to 512, the output is mapped to a latent variable of dimension 16. The decoder mirrors the encoder and maps the latent variable of dimension 16, via three layers, to the original input size.

For the residual architecture we present specific tables for the encoder (Table 4), decoder (Table 5), and the residual block (Table 6).

We train each VAE on a single NVIDIA GeForce RTX 2080 Ti 11GB GPU.

Table 4: Residual encoder architecture.

| layer | dimension | additional parameters |
|---|---|---|
| Convolution | $3 \to n_c$ | kernel size: 4, stride: 2, padding: 1, no bias |
| BatchNorm | $n_c$ | |
| ReLU | | |
| Convolution | $n_c \to n_c$ | kernel size: 4, stride: 2, padding: 1, no bias |
| BatchNorm | $n_c$ | |
| Residual | $n_c$ | |
| Residual | $n_c$ | |
| Convolution | $n_c \to 2d_z$ | kernel size: 1, stride: 1, padding: 0, bias |

Table 5: Residual decoder architecture.

| layer | dimension | additional parameters |
|---|---|---|
| Convolution | $d_z \to n_c$ | kernel size: 1, stride: 1, padding: 0, no bias |
| BatchNorm | $n_c$ | |
| Residual | $n_c$ | |
| Residual | $n_c$ | |
| Transposed Convolution | $n_c \to n_c$ | kernel size: 4, stride: 2, padding: 1, no bias |
| BatchNorm | $n_c$ | |
| ReLU | | |
| Convolution | $n_c \to 3$ | kernel size: 1, stride: 1, padding: 0, bias |

Table 6: Architecture of a residual block.

| layer | dimension | additional parameters |
|---|---|---|
| ReLU | | |
| Convolution | $n_c \to n_c$ | kernel size: 3, stride: 1, padding: 1, no bias |
| BatchNorm | $n_c$ | |
| ReLU | | |
| Convolution | $n_c \to n_c$ | kernel size: 3, stride: 1, padding: 1, no bias |
| BatchNorm | $n_c$ | |

## B.2  Reconstructions

Figure 9 shows reconstructions for CIFAR-10. We reconstruct a random subset of 9 samples from the test set. The VAEs we use are quite small and simple, they are also non-hierachical, which makes it hard to model CIFAR-10 well. Therefore, qualitatively the differences between the methods are quite subtle. But still, we can see that the DMaaPx's reconstruction is slightly better for the rightmost image in the second row.

## B.3  Additional Datasets

Figure 10 shows the three generalization gaps for the BinaryMNIST dataset. Figure 11 shows the three generalization gaps for the Fashion-MNIST dataset. All claims that have been made in the main text also hold for these datasets.

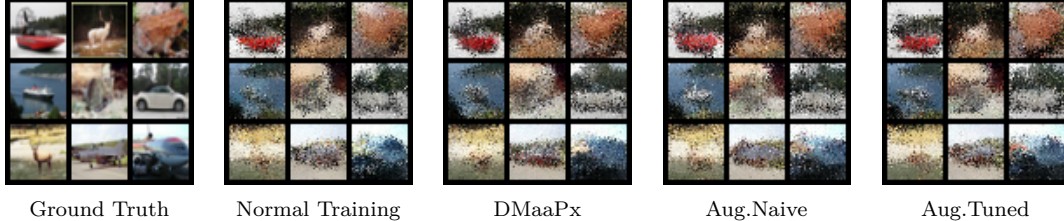

| Ground Truth | Normal Training | DMaaPx | Aug.Naive | Aug.Tuned |

Figure 9: Reconstructions for CIFAR-10 for a random subset of 9 samples from the test set.

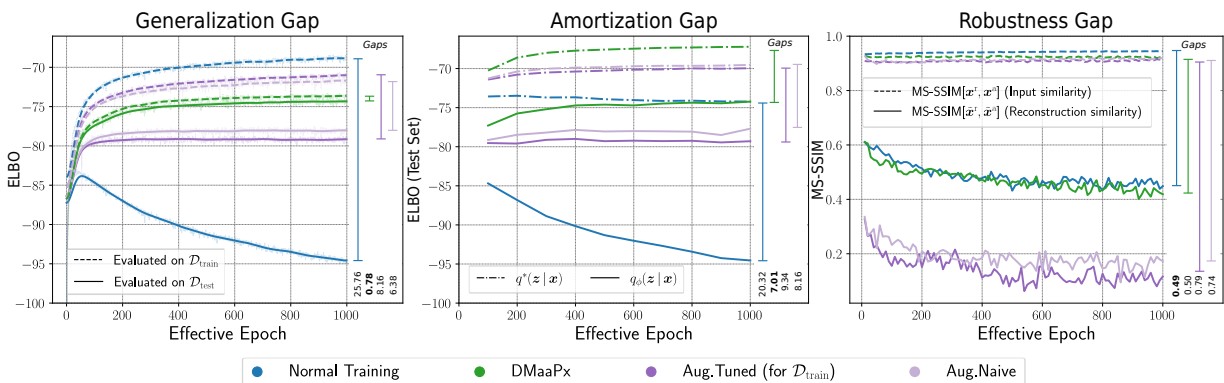

Figure 10: Generalization gap, amortization gap, and robustness gap on the BinaryMNIST dataset.

## B.4 Different Conditional Likelihoods

VAEs' modeling assumptions for the conditional likelihood $p_\theta(\boldsymbol{x} \mid \boldsymbol{z})$ often differ based on data or use case. While a Gaussian likelihood is used for applications that focus on low reconstruction error (e.g., lossy data compression), a MoL likelihood is used if the density of the data matters (e.g., generative modeling or lossless data compression). While the experiments in the main text considered a MoL likelihood, the model trained on FashionMNIST (Figure 11) uses a Gaussian likelihood and the model trained on BinaryMNIST (Figure 10) uses a Bernoulli likelihood (also compare Appendix B.1). Figure 12 also evaluates MoL for FashionMNIST, and we observe a similar behavior as in its Gaussian counterpart in Figure 11 (left). In summary, DMaaPx is less prone to overfitting than normal training and augmentation, for all investigated likelihoods.

## B.5 Training ELBO versus ELBO on $\mathcal{D}_{\text{train}}$

**Remark** (Test data entropy can also affect the ELBO value). Note that from Eqs. (2) and (1), we have

$$\mathbb{E}_{\boldsymbol{x} \sim p(\boldsymbol{x})}\left[\text{ELBO}_\Theta(\boldsymbol{x})\right] \leq \mathbb{E}_{\boldsymbol{x} \sim p(\boldsymbol{x})}\left[\log p_\theta(\boldsymbol{x})\right] = -H[p(\boldsymbol{x}), p_\theta(\boldsymbol{x})] \leq -H[p(\boldsymbol{x})], \tag{11}$$

where $H$ denotes the (cross) entropy. Therefore, the ELBO on $\mathcal{D}_{\text{test}}$ can be higher than the ELBO on $\mathcal{D}_{\text{train}}$, if $\mathcal{D}_{\text{train}}$ and $\mathcal{D}_{\text{test}}$ are not drawn from the same distribution, and $\mathcal{D}_{\text{test}}$ has a lower entropy than $\mathcal{D}_{\text{train}}$. Indeed, this phenomenon has been observed in the out-of-distribution setting when testing on a low-entropy data set (Nalisnick et al., 2018).

Figure 13 shows the ELBOs analogous to the generalization gaps in Figure 2, Figure 10, and Figure 11, but the dotted lines plot the ELBO on the actual training distribution (e.g., on samples from $p_{\text{DM}}(\boldsymbol{x}')$ for DMaaPx). The point of this plot is to warn that comparisons between ELBOs under such different distributions are not meaningful, and should not be used to calculate the generalization gap. For example, note that the plot would suggest a negative generalization gap for data augmentation (purple) on BinaryMNIST. This is consistent with the remark above: since the ELBO is bounded by the negative entropy of the distributions on

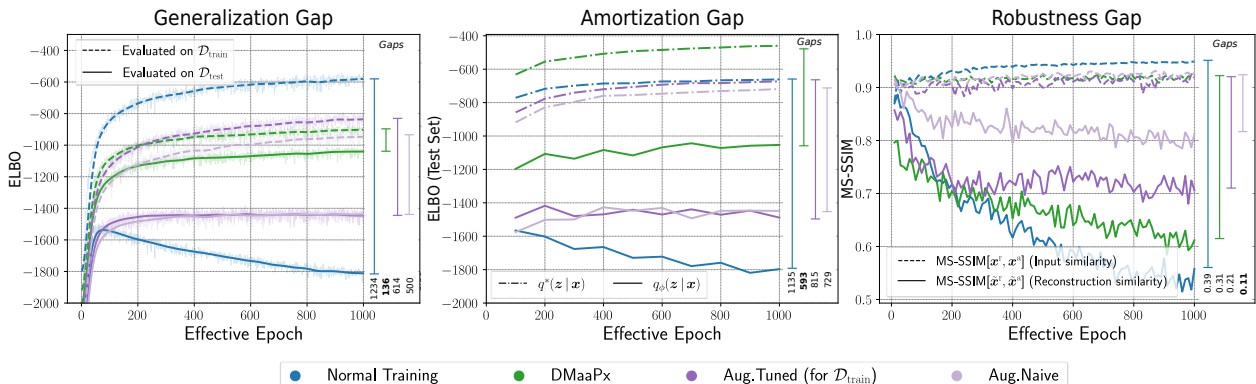

Figure 11: Generalization gap, amortization gap, and robustness gap on the FashionMNIST dataset.

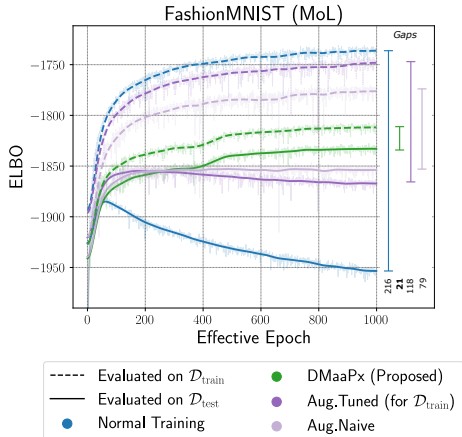

Figure 12: Generalization performance for FashionMINST with MoL likelihood. We observe similar behavior to left panel in Figure 11, which uses a Gaussian likelihood.

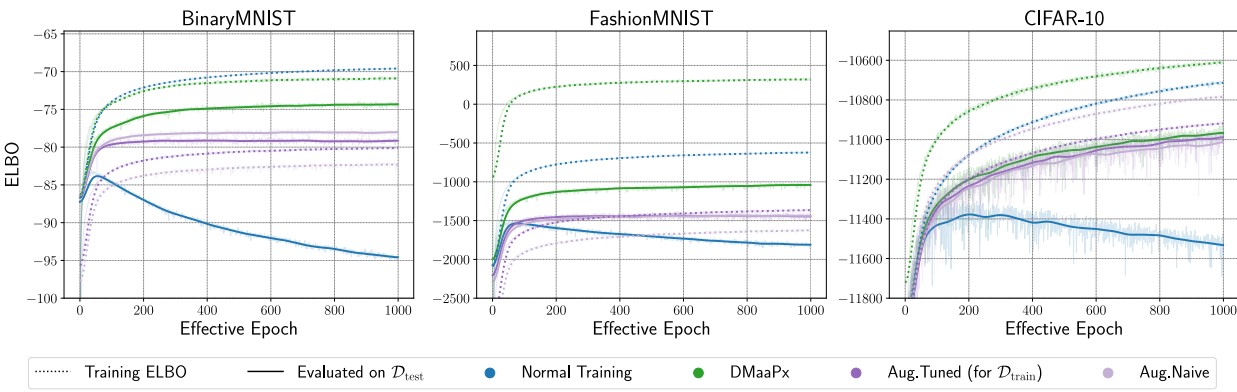

Figure 13: ELBO evaluted on the distribution that is actually used for training (dotted, see Eqs. (7)-(8)). For augmentations, the test ELBO (solid) is higher than the training ELBO (dotted) in the left two panels, which is an artifact of different entropies of the distributions, see Remark.

which it is evaluated, evaluating it on two different distributions with different entropies exhibits differences unrelated to the generalization gap.

## B.6 Adding Scheduled Noise to the Training Set

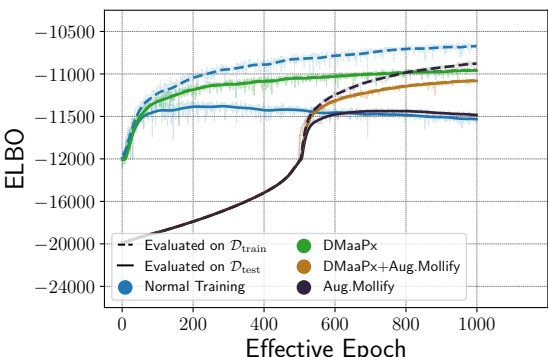

Figure 14: Generalization gap as a function of effective epochs. We compare normal training to DMaaPx, Aug.Mollify, and the combination of DMaaPx and Aug.Mollify. Aug.Mollify reaches a similar performance as the baseline. Combining DMaaPx and Aug.Mollify does not improve upon DMaaPx.

Tran et al. (Tran et al., 2023) show in their recent work that adding (scheduled) noise to the training data during the training process helps the optimization of likelihood-based generative models. More specifically, they anneal the noise schedule from a large variance to a small one for half of the training epochs and, finally, use the original training data for the other half of the training epochs. They argue that data typically resides on a low-dimensional manifold that is embedded in a larger dimension. By adding noise to the data during training one circumvents the problem of density estimation in input space with few samples and, additionally, prevents the model from overfitting to the data manifold. One can also view this idea from a continuation method perspective as the method finds a good initialization point for the optimization.

Figure 14 shows the performance of the mollification method (Aug.Mollify) applied to VAEs on the CIFAR-10 dataset (dark). We compare the performance to normal training (blue), DMaaPx (green), and the combination of DMaaPx and the Aug.Mollify. We find that Aug.Mollify reaches a similar performance as the baseline but does not improve upon it. Also, combining DMaaPx and Aug.Mollify does not bring improvements compared to DMaaPx. However, DMaaPx and Aug.Mollify have similarly small generalization gaps.

## B.7 Mixing Synthetic Data with Real Data

We evaluate the performance of VAEs trained on a mix of $x \in \{0, 20, 40, 60, 80, 100\}\%$ of real data and $100 - x\%$ of synthetic data. We train the models without augmentation and with Aug.Tuned. Note that $x = 0\%$ (without augmentation) corresponds to the DMaaPx method and $x = 100\%$ (without augmentation) corresponds to the normal training baseline. $x = 100\%$ (with augmentation) corresponds to the Aug.Tuned baseline.

Considering no augmentation, when moving from a mix of 100% of real data to 0% of real data, we find that the performance is continuously increasing (i.e., the ELBO evaluated on $\mathcal{D}_{\text{test}}$ increases) and the generalization gap is shrinking. Adding augmentation improves both, the ELBO evaluated on $\mathcal{D}_{\text{test}}$ and the generalization gap. Note that DMaaPx ($x = 0\%$, no augmentation) shows the highest ELBO evaluated on $\mathcal{D}_{\text{test}}$ and the smallest generalization gap.

## B.8 Training the Diffusion Model on Subsets of the Training Set

We evaluate the perfomance of DMaaPx trained on samples from a diffusion model which has been trained on $x \in \{10, 30, 50, 70, 90\}\%$ of CIFAR-10.

Figure 16 shows ELBO values evaluated on $\mathcal{D}_{\text{train}}$ and $\mathcal{D}_{\text{test}}$. The test performance of DMaaPx using samples from a diffusion model that has been trained on 30% of data equals the test performance of normal training.

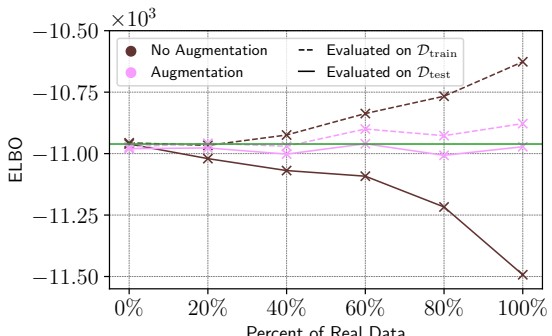

Figure 15: Last ELBO evaluated on $\mathcal{D}_{\text{train}}$ (dashed) and $\mathcal{D}_{\text{test}}$ (solid) with respect to $x\%$ of real data for no augmentation (brown) and Aug.Tuned (pink). $x = 0\%$, brown: DMaaPx. $x = 100\%$, brown: Normal Training. $x = 100\%$, pink: Aug.Tuned. The vertical line shows the ELBO evaluated on the test set for DMaaPx. DMaaPx has highest ELBO and smallest generalization gap. More details are provided in Appendix B.7.

However, DMaaPx shows a significantly smaller generalization gap. Using more data for training the diffusion model improves upon normal training with diminishing returns for $\geq 70\%$ of data.

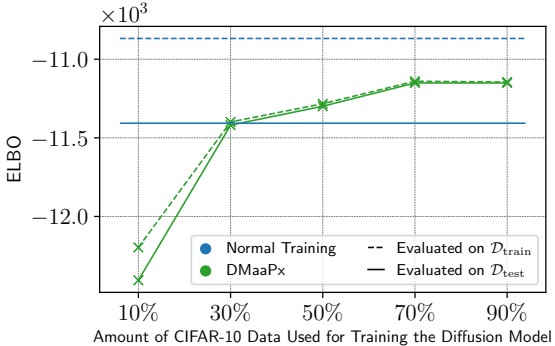

Figure 16: ELBO evaluated on $\mathcal{D}_{\text{train}}$ (dashed) and $\mathcal{D}_{\text{test}}$ (solid) for DMaaPx (green) trained on samples from a diffusion model which has been trained on $x \in \{10, 30, 50, 70, 90\}\%$ of CIFAR-10. We plot the performance of normal training (i.e., VAE trained with the original full CIFAR-10 only) as horizontal blue lines. Using 30% of data for DMaaPx shows similar performance than normal training (with a significantly smaller gap). Using more data outperforms normal training. More details are provided in Appendix B.8.

## B.9   Algorithms for Training VAEs with DMaaPx

Below, we provide pseudocode for training VAEs with normal training (Algorithm 1) and DMaaPx (Algorithm 2).

---

**Algorithm 1** Training VAEs - Normal

---

Given: $\mathcal{D}_{\text{train}}$, $\Theta_0 = \{\theta_0, \phi_0\}$, Number of epochs $T$, Batch size $M$

$j = 1$;
**for** $t = 1, \cdots, T$ **do**

    **for** $i = 1, \cdots, \frac{|\mathcal{D}_{\text{train}}|}{M}$ **do**

        Get $M$ training examples: $[\boldsymbol{x}_1, \cdots, \boldsymbol{x}_M] = \mathcal{D}_{\text{train}}[(i-1) \times M : i \times M]$;

        $\mathcal{L}([\boldsymbol{x}_1, \cdots, \boldsymbol{x}_M]; \Theta_{j-1}) = -\frac{1}{M} \sum_m [\text{ELBO}_{\Theta_{j-1}}(\boldsymbol{x}_m)]$;

        $\Theta_j = \Theta_{j-1} - \eta \cdot \nabla \mathcal{L}$;

        $j = j + 1$;

    **end**

**end**

---

**Algorithm 2** Training VAEs - DMaaPx

---

Given: $p_{\text{DM}}(\boldsymbol{x})$, $\Theta_0 = \{\theta_0, \phi_0\}$, Number of epochs $T$, Batch size $M$, $N = |\mathcal{D}_{\text{train}}|$

**// Step 1: Sampling the training data**

$\mathcal{D}_{\text{DM}} = [\boldsymbol{x}_1, \cdots, \boldsymbol{x}_{T \times N}] \sim p_{\text{DM}}(\boldsymbol{x})$    // Parallelizable; only need to do this once.

**// Step 2: Training the VAE**

$j = 1$;
**for** $t = 1, \cdots, T$ **do**

    **for** $i = 1, \cdots, \frac{N}{M}$ **do**

        Get $M$ training examples: $[\boldsymbol{x}_1, \cdots, \boldsymbol{x}_M] = \mathcal{D}_{\text{DM}}[t \times (i-1) \times M : t \times i \times M]$;

        $\mathcal{L}([\boldsymbol{x}_1, \cdots, \boldsymbol{x}_M]; \Theta_{j-1}) = -\frac{1}{M} \sum_m [\text{ELBO}_{\Theta_{j-1}}(\boldsymbol{x}_m)]$;

        $\Theta_j = \Theta_{j-1} - \eta \cdot \nabla \mathcal{L}$

        $j = j + 1$;

    **end**

**end**

---

## C  Quantitative Results on Performance Gaps

Table 7 assigns quantitative values to the visual evidence in Figure 2 (CIFAR-10), Figure 11 (FashionMNIST), and Figure 10 (BinaryMNIST).

Table 7:  Quantitative values of the performance gaps visualized in Figure 2, Figure 11, and Figure 10. Bold numbers indicate the smallest gap within a dataset.

| | | generalization gap $(\mathcal{G}_{\mathrm{g}}$, Eq. (3)) | amorization gap $(\mathcal{G}_{\mathrm{a}}$ , Eq. (4)) | robustness gap $(\mathcal{G}_{\mathrm{r}}$, Eq. (5)) |
|---|---|---|---|---|
| Binary MNIST | Normal Training | 25.76 | 20.32 | **0.49** |
| | DMaaPx (ours) | **0.78** | **7.01** | 0.50 |
| | Aug.Tuned | 8.16 | 9.34 | 0.79 |
| | Aug.Naive | 6.38 | 8.16 | 0.74 |
| Fashion MNIST | Normal Training | 1234.50 | 1135.89 | 0.39 |
| | DMaaPx (ours) | **136.57** | **593.39** | 0.31 |
| | Aug.Tuned | 614.93 | 815.52 | 0.21 |
| | Aug.Naive | 500.33 | 729.83 | **0.11** |
| CIFAR-10 | Normal Training | 841.54 | 835.86 | 0.41 |
| | DMaaPx (ours) | **5.44** | **288.82** | **0.30** |
| | Aug.Tuned | 94.28 | 328.08 | 0.35 |
| | Aug.Naive | 228.05 | 390.25 | 0.35 |

# D    Diffusion Model for DMaaPx

We follow the setup of Karras et al. (Karras et al., 2022) for the design and training of our diffusion model. However, we do not use the proposed augmentation pipeline during training.

We train the diffusion model on 50,000 training data points (i.e., the size of the original CIFAR-10 training set in the case of CIFAR-10) for 4000 epochs. Each model is trained on 8 NVIDIA A100 80GB GPUs for approximately 1 days.

We utilized the deterministic second-order sampler as proposed by Karras et al. (Karras et al., 2022) with 18 integration steps. Each sampled image utilizes a unique initial seed. We sample on a single NVIDIA A100 40GB GPU. Sampling 50,000 images takes approximately 25 to 30 minutes.

## D.1    Samples From the Diffusion Model

Figure 17 shows samples from the diffusion models trained. On CIFAR-10 we report a FID score of 3.9537. Scores on BinaryMNIST and FashionMNIST are ommited as those are not widely reported and heavily depend on preprocessing (Song et al., 2021).

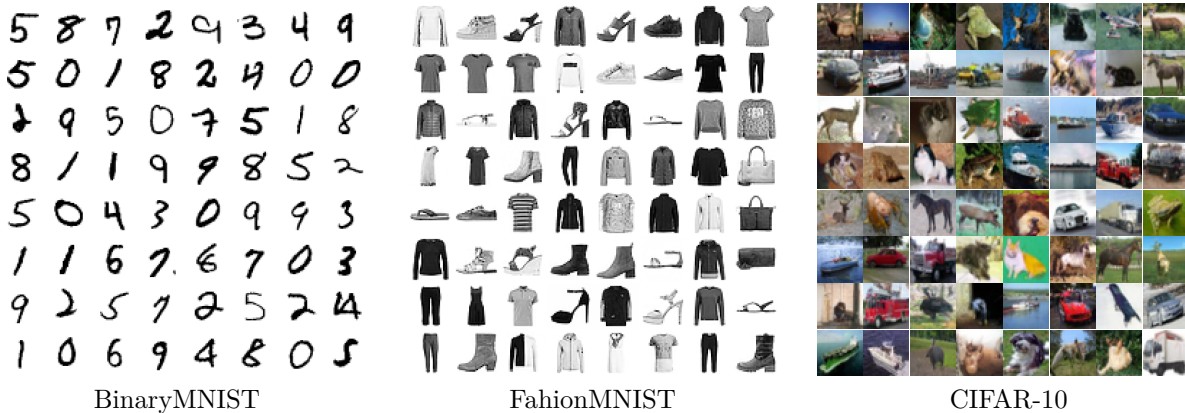

BinaryMNIST                    FahionMNIST                    CIFAR-10

Figure 17: Samples of the diffusion models trained on BinaryMNIST (LeCun et al., 1998), FashionMNIST (Xiao et al., 2017), and CIFAR-10 (Krizhevsky et al., 2009).

## D.2    Computational Cost of DMaaPx

This section discusses the computational requirements of the DMaaPx method in general.

In this paper, we use the method DMaaPx as a tool to study the difference between naive augmentations and synthetic data from diffusion models when training VAEs. Our evaluations show that there are scenarios where exploiting the modelling assumption (i.e., both VAEs and diffusion models are probabilistic models) helps, which is often overlooked by the generative modelling community.

If a pre-trained diffusion model is available, DMaaPx can be applied with a negligible computational overhead (see Figure 4, right). This setup follows recent trends where using synthetic data from pre-trained diffusion models is becoming a common practice for, e.g., classification (Azizi et al., 2023), robustness (Croce et al., 2021; Wang et al., 2023), and representation learning (Tian et al., 2023).

If no pre-trained diffusion model is available, a diffusion model has to be trained in a first step to apply DMaaPx. Whether this is feasible might depend on the specific application. However, we want to emphasize that once the diffusion model is trained, it can be used to generate as many samples as needed for training as many VAEs as needed.

### D.3    Discriminating Synthetic and Real Samples by Classification

One way to evaluate the quality of the generated samples is by training a classifier to discriminate the synthetic and the real data. This method is also known as Classifier Two-Sample Tests (C2ST), which is a metric for evaluating generative models (Bischoff et al., 2024). Here, we train five classifiers to discriminate real data (images taken from CIFAR-10) and synthetic data (images sampled from the diffusion models) where we use $x \in \{30, 50, 70, 90, 100\}\%$ of the original CIFAR-10 to train the diffusion models, respectively.

We use a ResNet 18 (He et al., 2016) for classifying real data (with label 1) and synthetic data (generated by the diffusion model). We train each classifier for 14 epochs with a batch size of 256 with stochastic gradient descent (Robbins & Monro, 1951) with a learning rate of 0.001, a momentum (Rumelhart et al., 1986) of 0.9 and weight decay of $5 \times 10^{-4}$.

Figure 18 shows the classification accuracies for discriminating between real and synthetic data. While all samples can be classified with an accuracy $> 50\%$, we find a negative trend between the amount of data that the diffusion model is trained on ($x \in \{30, 50, 70, 90, 100\}\%$) and the classification accuracy. Hence, it is easier to classify samples from a diffusion model that has been trained on less data than samples from a diffusion model that has been trained on more data. Using the maximum amount of training data available, which DMaaPx does, leads to samples that can be distinguished least from real data.

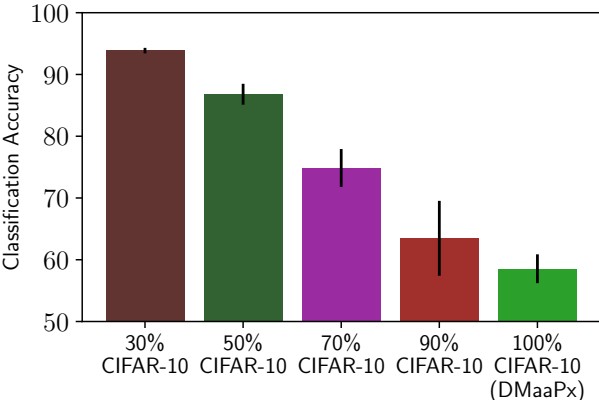

Figure 18:    Accuracy of classifier trained to distinguish real data (CIFAR-10) and data sampled from a diffusion model. The diffusion model is trained on $x \in \{30, 50, 70, 90, 100\}\%$ of real data. The more data is used the closer synthetic samples are to real samples (i.e., smaller classification accuracy). Using 100% of CIFAR-10 leads to samples least distinguishable from real data. Accuracies are averaged over 5 random seeds. Error bars show $\pm 1.96$ standard deviations. For details see Appendix D.3.

# E    Augmentation

We use the augmentation pipeline originally proposed for GAN training following Karras et al. (Karras et al., 2020). Each specific augmentation is applied with a probability of $b \in \{0.1, 0.12\}$. For each dataset we compare two sets of specific augmentations.

1. The hyperparameters for each specific augmentation are tuned by hand with the goal of imitating the data generating distribution that produced the dataset.

2. We use a naive set of specific augmentations that is targeted to image datasets (in general).

Table 8 lists naive augmentation for BinaryMNIST, FashionMNIST, and CIFAR-10. Table 9 lists augmentation tuned to the BinaryMNIST and the FashionMNIST dataset. Table 10 lists augmentation tuned to the CIFAR-10 datset.

They perform similarly overall in Figures 2, 10 and 11. However, Aug.Naive outperforms Aug.Tuned in generalization on BinaryMNIST and FashionMNIST, and in robustness across all datasets. This is surprising as naive augmentation might produce out-of-distribution data, like a horizontally flipped digit "2", potentially impairing performance. Thus, designing augmentation can be labor-intensive.

Table 8:  List of specific augmentations applied to BinaryMNIST, FashionMNIST and CIFAR-10. We refer to this set as "naive" augmentation as it is targeted towards images in general (and not towards specific datasets). Each specific augmentation is applied with probability $b$.

| augmentation | description and hyperparameters |
|---|---|
| horizontal flip | flip an image horizontally |
| translation | translate an image in $x$ and $y$ direction for $t \in \{0, 1, 2, 3\}$ pixels |
| scaling | scale an image by $2^{\sigma_{\mathrm{scale}}}$ with $\sigma_{\mathrm{scale}} \in [0, 0.2]$ |
| rotation | rotate an image by $d$ degrees with $d \in [0, 10]$ |
| anisotropic scaling | do anisotropic scaling with scale $2^{\sigma_{\mathrm{aniso-scale}}}$ ($\sigma_{\mathrm{aniso-scale}} \in [0, 0.2]$) |
| anisotropic rotation | do anisotropic rotation with a probability of 0.5 |
| brightness | change the brightness of an image by $\sigma_{\mathrm{brightness}} \in [0, 0.2]$ |
| contrast | change the contrast of an image by $2^{\sigma_{\mathrm{contrast}}}$ where $\sigma_{\mathrm{contrast}} \in [0, 0.25]$ |
| hue | change the hue by rotation of $r_{\mathrm{hue}}$ with $r_{\mathrm{hue}} \in [0, 0.25 \cdot \pi]$ |
| saturation | change the saturation of an image by $2^{\sigma_{\mathrm{saturation}}}$ where $\sigma_{\mathrm{saturation}} \in [0, 0.5]$ |

Table 9:  List of specific augmentations applied to BinaryMNIST and FashionMNIST. The set is tuned towards BinaryMNIST and FashionMNIST. Each specific augmentation is applied with probability $b$.

| augmentation | description and hyperparameters |
|---|---|
| translation | translate an image in $x$ and $y$ direction for $t \in \{0, 1, 2, 3\}$ pixels |
| scaling | scale an image by $2^{\sigma_{\mathrm{scale}}}$ with $\sigma_{\mathrm{scale}} \in [0, 0.15]$ |
| rotation | rotate an image by $d$ degrees with $d \in [0, 10]$ |
| anisotropic scaling | do anisotropic scaling with scale $2^{\sigma_{\mathrm{aniso-scale}}}$ ($\sigma_{\mathrm{aniso-scale}} \in [0, 0.15]$) |
| anisotropic rotation | do anisotropic rotation with a probability of 0.4 |

Table 10: List of specific augmentations applied to CIFAR-10. The set is tuned towards CIFAR-10. Each specific augmentation is applied with probability $b$.

| augmentation | description and hyperparameters |
|---|---|
| horizontal flip | flip an image horizontally (applied with probability 1) |
| vertical flip | flip an image vertically |
| scaling | scale an image by $2^{\sigma_{\text{scale}}}$ with $\sigma_{\text{scale}} \in [0, 0.2]$ |
| rotation | rotate an image by $d$ degrees with $d \in [0, 360]$ |
| anisotropic scaling | do anisotropic scaling with scale $2^{\sigma_{\text{aniso}-\text{scale}}}$ ($\sigma_{\text{aniso}-\text{scale}} \in [0, 0.2]$) |
| anisotropic rotation | do anisotropic rotation with a probability of 0.5 |

# F   Practical Evaluation of VAEs on Three Tasks

The improvements of generalization performance, amortized inference and robustness in VAEs have direct impacts on the applications of them. In this section, we evaluate three popular tasks that VAEs are used for, based on whether a task only involves the encoder, the decoder, or both as in (Xiao & Bamler, 2023): (a) representation learning (i.e., using only the encoder); (b) data reconstruction (i.e., using both the encoder and the decoder); and (c) sample generation (i.e., using only the decoder).

**Representation learning (with classification as the downstream task).**   We evaluate the representation learning performance by classification accuracies on the mean $\boldsymbol{\mu}$ of $q_\phi(\boldsymbol{z} \,|\, \boldsymbol{x})$ for each $\boldsymbol{x}$. First, we find the learned representations $\boldsymbol{\mu}$ for all data points in the CIFAR-10 test set. Afterwards, we split them into two separate subsets. We use one subset to train the classifier, and test it on the other subset. Our experiments include four different classifiers: logistic regression, a support vector machine (Boser et al., 1992) with radial basis function kernel (SVM-RBF), a SVM with linear kernel (SVM-L), and $k$-nearest neighbors (kNN) with $k = 5$. Table 11 (representation learning; **RL**) shows the resulting test accuracies across all models considered. We find that VAEs trained with DMaaPx (in bold) are slightly better than other models on average (with overlapping standard deviations), which implies the task of representation learning might benefits from the smaller gaps evaluated in Section 5.2.

**Data reconstruction.**   Tasks such as lossy data compression (Ballé et al., 2017) rely on the reconstruction performance of VAEs. We evaluate the reconstruction performance of VAEs trained on CIFAR-10 using the peak signal-to-noise ratio (PSNR; higher is better). Table 11 (reconstruction; **RC**) shows that DMaaPx performs slightly better than the others on average, but with overlapping standard deviations.

**Sample generation.**   We evaluate the quality of samples generated by VAEs trained on CIFAR-10 with the methods explained in the main text (Normal Training, DMaaPx, Aug.Naive, Aug.Tuned). We report Fréchet Inception Distance (Heusel et al., 2017) (FID; lower is better) and Inception Score (Salimans et al., 2018) (IS; higher is better). Table 11 (sample quality; **SQ**) shows that DMaaPx performs slightly better than the others on average, with overlapping standard deviations, when sample quality is measured in FID, but Normal Training performs better when sample quality is measured in IS.

Overall, VAEs trained with DMaaPx show improvements for representation learning and data reconstruction, and perform similarly to normal training on sample quality. At the same time, VAEs trained with both augmentations seem to have slightly worse performance for representation learning and sample generation, and perform similarly on the reconstruction task when compared to normal training. The results of DMaaPx in the table is consistent with our claim that the proposed method mainly fixes the encoder, which affects representation learning and reconstruction but not sample quality. Additionally, Theis et al. (Theis et al., 2016) show that a generative model with good log-likelihood (i.e., high test ELBO in the case of a VAE) does not necessarily produce great samples.

Table 11:   Evaluation of downstream applications of VAEs on CIFAR-10: representation learning with classification as the downstream task (**RL**), reconstruction (**RC**), and sample quality (**SQ**). Results are averaged over 3 random seeds. Note that most differences are smaller than the standard deviations. See Appendix F for a discussion of the results.

|  |  | Normal Training | DMaaPx (ours) | Aug.Naive | Aug.Tuned |
|---|---|---|---|---|---|
| **RL** | log. reg. (↑) | $0.370 \pm 0.018$ | $\mathbf{0.383 \pm 0.018}$ | $0.359 \pm 0.004$ | $0.361 \pm 0.014$ |
|  | SVM-RBF (↑) | $0.427 \pm 0.014$ | $\mathbf{0.438 \pm 0.015}$ | $0.421 \pm 0.004$ | $0.420 \pm 0.016$ |
|  | SVM-L (↑) | $0.367 \pm 0.015$ | $\mathbf{0.380 \pm 0.014}$ | $0.365 \pm 0.005$ | $0.366 \pm 0.022$ |
|  | kNN (↑) | $0.325 \pm 0.006$ | $\mathbf{0.327 \pm 0.035}$ | $0.300 \pm 0.004$ | $0.299 \pm 0.028$ |
| **RC** | PSNR (↑) | $16.087 \pm 0.042$ | $\mathbf{16.370 \pm 0.195}$ | $16.105 \pm 0.017$ | $15.924 \pm 0.205$ |
| **SQ** | FID (↓) | $219.256 \pm 16.124$ | $\mathbf{219.081 \pm 14.894}$ | $237.238 \pm 43.218$ | $240.898 \pm 11.072$ |
|  | IS (↑) | $\mathbf{1.818 \pm 0.155}$ | $1.614 \pm 0.076$ | $1.656 \pm 0.047$ | $1.612 \pm 0.083$ |

## G  Practical Evaluation of VAEs on CIFAR-10-C

On CIFAR-10-C (Hendrycks & Dietterich, 2019), we evaluate the representation learning capabilities of VAEs trained with DMaaPx, Aug.Naive, and Aug.Tuned by evaluating the classification performance of a classifier trained in latent space. The experiment are similar to the evaluation of representation learning above (see Appendix F). For each of the 19 classes of corruptions we train four classifiers on a subset of the CIFAR-10-C test set and evaluate its performance on the subset that was not used for training (termed "overall" performance). Additionally, we sample a random (one out of 19 possible) corruption for each of the images of CIFAR-10-C and evaluate its performance. During training we use the strongest degree of corruption.

Figure 19 shows results across all 19 corruptions and the "overall" performance. DMaaPx achieves consistently the best mean across all corruptions for the linear classifiers and DMaaPx achieves the best mean for almost all of the corruptions when regarding the non-linear classifiers. See also Section 5.2 for a discussion of the experiments.

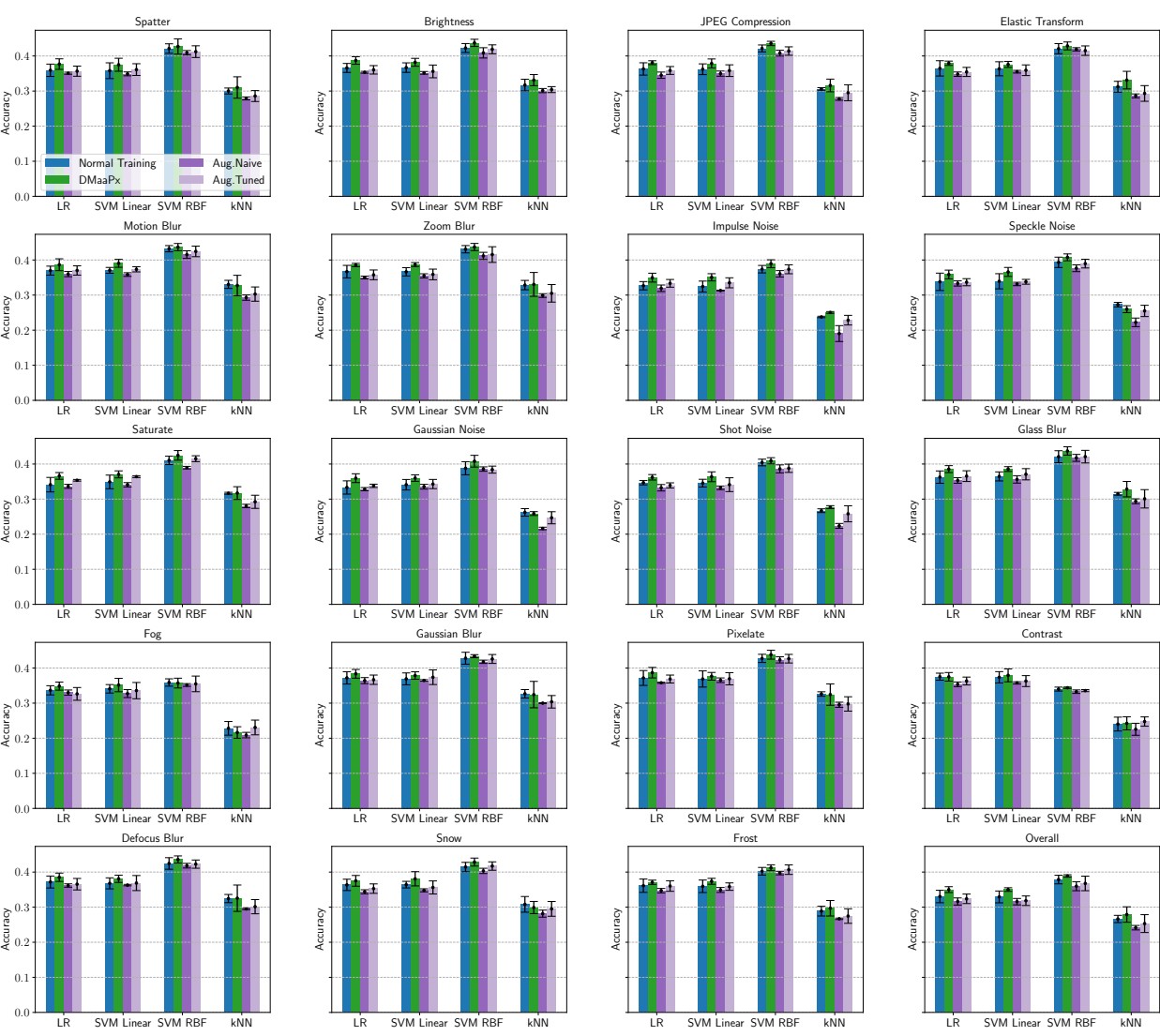

Figure 19: Classification performance of four classifiers (LR: linear regression, SVM kernel: support vector machine with the specified kernel, kNN: $k$-nearest-neighbor classifier) trained in latent space across the 19 corruptions that are part of CIFAR-10-C. DMaaPx achieves the best mean performance for almost all corruptions under consideration. For details consult Appendix G.

