# OpenReview forum: "A Note on Generalization in Variational Autoencoders: How Effective Is Synthetic Data and Overparameterization?"
_TMLR — Accepted by TMLR_

### Review · Reviewer_tiKc · 2024-11-04

**Summary Of Contributions:**

This paper focuses on the effectiveness on synthetic data and over-parametrization on variational autoencoder training. The authors introduce a method coined as DMaaPx which consists in training a generative diffusion model over the training data distribution and then use it to sample a potential infinite training dataset for training a variational autoencoder. The authors compare DMaaPx with traditional handcrafted data augmentation technique and show that DMaaPx gives better results than their methods on the following metric Generalization Metric, Amortization Gap and Robustness GAP that are based on the ELBO. The authors also state that DMaaPx is outperforming baselines on downstream and generation tasks. They also study the relationship between the training data and the model's capacity, the authors give some guidelines on how to increase the model's capacity without hurting generalization.

**Audience:**

Yes

**Claims And Evidence:**

Yes

**Requested Changes:**

- Can you clarify that the VAE when using DMaaPx is trained only on synthetic generate data and not on a mix of real/synthetic? From the paper, my understanding is that the authors are only training on synthetic data. In any case, I think it would add strength to the paper to add some ablation over mixing real with synthetic data (with and without additional data augmentation).
- One point that often come back when using synthetic data from diffusion models is that despite having visually appealing sample, it's still easy for a model to learn to discriminate between real and generate data. Since one of the assumption made in the paper is that the diffusion model is able to provide an accurate approximation of D_train, I would have expected a more in depth analysis on wether this is the case and what are the limitations. In other work, what would happen if we train a simple model is discriminating between fake and real sample? If we can learn a model that could learn that, would that still mean that we are accurately approximating d_train?
- It it possible to add samples in the appendix from the trained VAE? Would be great to have some reconstruction along table 11.
- Can you downplay the claims around DMaaPx outperforming other methods on downstream tasks?

Overall, I think this paper can have value for the community, but instead of focusing only on DMaaPx outperforming other tasks, it would be better to recenter the paper on achieving competitive performances using only synthetic data which is already a great contribution.

**Strengths And Weaknesses:**

Strengths:
- The paper is extremely well-written, clear to follow.
- The explanations are great, I really like Figure 1. It's also great to have the focus both on the data and also on the model capacity.
 - I appreciate the details in the appendix.

Weaknesses:
- A shortcoming of this work is the motivation. The authors are training an unsupervised model to train an unsupervised model. There are some works around learning latent variable directly through diffusion models. So it's not clear why someone should use the method's presented by the author instead of training a single diffusion model. On a similar line, the paper would have been much stronger is the authors were able to run similar evaluations as they did on the VAE directly on their diffusion model (several papers show that diffusion models can be seen as classifier and are able to solve many downstream tasks).
- Lack of analysis over the quality of the latent learned by DMaaPx. I would have been curious to know wether changing the data distribution between real and synthetic could change some properties in the latent space that is learned. I would expect that since the distribution learned by the diffusion model should be smoother, it might also be easier to interpolate in the VAE latent space.
- Another shortcoming is the over-reliance on the ELBO as metric. As noted by the authors, Nalisnick et al., 2018, show that ELBO on test can be higher than the one on the training data. And several works have highlighted the limitations over such metric. So, it would have been fine in the main paper to have maybe one figure with the ELBO but since all figures in the main paper rely on the ELBO, I have some concerned about the reliability of the plots. On a following point, the most interesting results are probably the ones in Table 11 in the appendix. For this figure, we have a bunch of downstream tasks such as classifications ones and generative ones. For the generative ones, the number between using real and generated data are very close (and FID, IS are not very reliable metric so, it doesn't seem that there is any statistically significant differences between those two methods to say that one is better than the other ones). For the classification tasks, it seems that the difference between DMaaPx and traditional training data is 0.01, which I do not think this is significant enough for claiming any "outperforming" statement. If we look at the std, there is a clear overlap between normal training and DMaaPx. So one could learn a model than could be better using normal data than DMaaPx.

---

> ### Author Response · Authors · 2024-11-29
> **Response from the authors (Part 1)**
>
> > **Q1: There are some works around learning latent variables directly through diffusion models. So it's not clear why someone should use the method presented by the author instead of training a single diffusion model.**
>
>
>
> **A1:**
> The line of work that tries to use DMs’ latent space focuses on using the activation values from the U-Net in the DMs (Kwon et al., 2022; Hahm et al., 2024). This often needs an auxiliary architecture and requires further training on top of a pre-trained model. Therefore, unlike for VAEs, it does not come out-of-the-box. Moreover, this line of work is not adopted by practitioners in many areas where VAEs are still the dominating tools for representation learning.
>
> Different to, e.g., activations in hidden layers of the learnt score function in a DM, the latent variables in a VAE are explicitly constrained with controllable strength by a so-called variational information bottleneck (Alemi et al., 2017).  This forces the VAE to remove any redundant information from the latents, making them more interpretable (Higgins et al, 2017). Hence, at the moment, VAEs are more widely used than diffusion models in natural sciences (e.g., for anomaly detection in the LHC at CERN (Cerri et al., 2019), and for learning interpretable representations in quantum optics (Flam-Shepherd et al., 2022) and neuroscience (Zhou et al., 2020; Gondur et al., 2023)). VAEs are also the dominating models in neural data compression due to their connection to rate-distortion theory (Yang et al., 2023).
>
> We do have a concise version of the explanation above in the paper page 2 paragraph 2, but we can further elaborate in details in the paper if this is helpful.
>
> ---
>
> Kwon et al. "Diffusion models already have a semantic latent space." 2022.
>
> Hahm et al. "Isometric Representation Learning for Disentangled Latent Space of Diffusion Models." 2024.
>
> Alemi et al. "Deep Variational Information Bottleneck." 2022.
>
> Higgins et al. “Beta-VAE: Learning basic visual concepts with a constrained variational framework.” 2017.
>
> Cerri et al. “Variational autoencoders for new physics mining at the Large Hadron Collider.” 2019.
>
> Flam-Shepherd et al. “Learning interpretable representations of entanglement in quantum optics experiments using deep generative models.” 2022.
>
> Zhou et al. "Learning identifiable and interpretable latent models of high-dimensional neural activity using pi-VAE." 2020.
>
> Gondur et al. "Multi-modal Gaussian Process Variational Autoencoders for Neural and Behavioral Data." 2024.
>
> Yang et al. “An introduction to neural data compression.” 2023.
>
> ---
>
> > **Q2: Whether changing the data distribution between real and synthetic could change some properties in the latent space that is learned. I would expect the distribution learned by the diffusion model should be smoother as well as interpolations in the VAE latent space.**
>
> **A2:**
> This is a very interesting comment! We did not think about checking the interpolations before. We **added B.6 Smoothness of Latent Space** into the appendix for the results. In the setting of CIFAR-10, we chose two data samples $x_0$ and $x_1$ and map them to latent space (i.e., $z_0$ and $z_1$) using either the VAE with normal training or the VAE with DMaaPx. Then, we linearly interpolate between them and evaluate the corresponding densities. The results are plotted in Figure 13, which shows that the densities of the interpolations in the DMaaPx trained latent space are generally higher and indeed much smoother than the normally trained VAE.

---

> ### Author Response · Authors · 2024-11-29
> **Response from the authors (Part 2)**
>
> > **Q3: over-reliance on the ELBO as a metric. As noted by the authors, Nalisnick et al., 2018, show that ELBO on test can be higher than the one on the training data.**
>
>
> **A3:**
> First, we want to highlight that the statement from (Nalisnick et al., 2018) should not be taken out of the specific conditions. As we have also discussed in Appendix **B.5 Training ELBO versus ELBO on $D_{train}$** the ELBO on $D_{test}$ can be higher than the ELBO on $D_{train}$ if $D_{train}$ and $D_{test}$ are not drawn from the same distribution, and $D_{test}$ has a lower entropy than $D_{train}$. This is particularly a problem when VAEs are used for out-of-distribution detection, which is also how Nalisnick et al. motivated their work! However, in our case, the ELBOs are always evaluated on the same $D_{train}$ (i.e., not on the augmented data or the synthetic data) and $D_{test}$ across training methods, which we know are indeed from the same distribution.
>
> In cases that $D_{train}$ and $D_{test}$ are from the same distribution, these ELBOs are good metrics in the sense that a small gap between them is a necessary criterion for good generalization performance, and, as long as $D_{train}$ and $D_{test}$ come from the same data generative distribution, the most natural definition of generalization performance is having a low gap between the ELBOs of $D_{train}$ and $D_{test}$.
>
> Additionally, our analysis also focuses on the trend of the ELBO during training. For example, in Figure 2 (left), we identify overfitting by the fact that the test ELBO decreases. This is motivated because the ELBO is the objective function of the optimization, not because of its probabilistic interpretation (similar to how, e.g., in training classifiers, a decreasing test accuracy would indicate overfitting).
>
> ---
>
> > **Q4: For Table 11 in the appendix, I do not think this is significant enough for claiming any "outperforming" statement. Can you downplay the claims around DMaaPx outperforming other methods on downstream tasks?**
>
> **A4:**
> Thanks for the suggestion! We have **updated** the paper and **downplayed** the statement from “outperforming” to “slightly better on average but with overlapping standard deviations” in Appendix F to better describe the results in Table 11.
>
> ---
>
> > **Q5: Can you clarify that the VAE when using DMaaPx is trained only on synthetic generated data and not on a mix of real/synthetic?**
>
>
> **A5:**
> Yes, in the case of DMaaPx, VAEs are only trained on synthetic data.
>
> ---
>
> > **Q6:  I think it would add strength to the paper to add some ablation over mixing real with synthetic data (with and without additional data augmentation).**
>
>
> **A6:**
> Great idea! We **added** this ablation study **B.8 Mixing Synthetic Data with Real Data** to the Appendix in the updated version of the paper. We evaluate the performance of VAEs trained with {0%, 20%, 40%, 60%, 80%, 100%} of real data and 100-n% of synthetic data. On top of the mixed dataset, we also tried with and without augmentation (i.e., Aug.Tuned). Figure 15 shows the results. Without augmentation, we can see that mixing even 20% of real data will lead to worse test set performance and enlarge the generalization gap. As for augmentation, if we augment the synthetic data only (i.e., Augmentation + DMaaPx), we don’t see any additional benefit. If we augment the mixed data, we see that the test set performance stays almost the same regardless of the proportion of the real data, but the generalization gap does increase as more real data is used. Overall, we don’t see too much benefit of mixing real data with synthetic data if we can generate as many samples as we need. Similarly, adding augmentation on top of synthetic data does not bring additional gain.

---

> ### Author Response · Authors · 2024-11-29
> **Response from the authors (Part 3)**
>
> > **Q7: One of the assumptions made in the paper is that the diffusion model is able to provide an accurate approximation of $p_{data}(x)$. What would happen if we train a simple model discriminating between synthetic and real data?**
>
>
> **A7:**
> Good idea! Actually, this is known as Classifier Two-Sample Tests (C2ST), which is an existing metric for generative models (Bischoff et al., 2024)! Out of curiosity, we did some new experiments based on this idea and **added Section D.3 Discriminating Synthetic and Real Samples by Classification** in the Appendix showing the results. We trained five diffusion models (DMs) with {30%, 50%, 70%, 90%, 100%} of the original CIFAR-10 training data. We know that the DMs trained with more data should approximate the $p_{data}(x)$ better. We then generated samples from these DMs, and trained five classifiers to discriminate the synthetic and real data. The resulting classification test accuracies are shown in Figure 18. We can see that, indeed, for DMs trained with more data, the test accuracy for discriminating between synthetic and real data dropped. The DM that was trained with 100% training data (i.e., used for DMaaPx) achieved around 56% test accuracy. This is still a bit off from random guessing (i.e., 50%), hence the classifier did learn something. Therefore, the diffusion model we used is not perfect, but it is among the best we can get at the time, and has been well recognized by the community (Karras et al., 2022).
>
> However, the goal of our paper does not contain showing whether the DM we used in the paper is perfectly approximating $p_{data}(x)$. We want to highlight the key takeaway here is that the more accurate a DM is in approximating $p_{data}(x)$, the better DMaaPx can be (see the newly added **B.9** in the Appendix for the corresponding DMaaPx performance). And, the current state-of-the-art DM is already showing the usefulness of DMaaPx. It is very likely there will be better generative models in the future, which are more accurate in approximating $p_{data}(x)$ by some metrics. By then, we can simply replace the diffusion model with such new generative models, and the method DMaaPx will still work due to the arguments in Section 4.1.
>
> ---
>
> Bischoff et al., “A Practical Guide to Sample-based Statistical Distances for Evaluating Generative Models in Science”, 2024
>
> ---
>
> > **Q8: Would be great to have some reconstruction along table 11.**
>
> **A8:**
> Sure! We **added** Figure 8 in the Appendix **B.2 Reconstructions** showing reconstructions from the four VAEs in Table 11. The VAEs we used are quite small and simple, they are also non-hierachical, which makes it hard to model CIFAR-10 well. Therefore, qualitatively the differences are quite subtle. But still, we can see that, for example the second row third image from the left (i.e., a car), DMaaPx’s reconstruction is slightly better.

---

> ### Comment · Reviewer_tiKc · 2024-12-18
>
> Thank you for the rebuttal and for answering my concerns. I really like the new revision of the paper. I do not have further comments.

---

### Review · Reviewer_aqGx · 2024-11-05

**Summary Of Contributions:**

In this work, the authors investigate overfitting when training a VAE from a "deep learning perspective". To that end, they propose first a new procedure called "DMaaPx" which aims to prevent as much as possible overfitting when training a VAE. The main idea is to use a pre-trained diffusion model, trained on the available dataset, which provides an accurate approximation of the real distribution $p_{data}$ and allows to draw as many samples as we want according to the resulting distribution $p_{DM}$. They show on some benchmark datasets that the new "DMaaPx" procedure improves generalization compared to a classical VAE. They second study the influence of the size of the model on overfitting, and show that increasing the size of the layers next to the latent space improves the generalization on test data.

**Audience:**

Yes

**Claims And Evidence:**

Yes

**Requested Changes:**

- See the "Weaknesses" and the "Questions and perspectives" sections.
- In Section 2, some notations are quite confusing. In particular, for clarity, I would appreciate to change $p(\mathbf{z})$ to $p_{\mathbf{Z}}(\mathbf{z})$ in order to clearly distinguish the prior distribution to the one of the data.

**Strengths And Weaknesses:**

**Strengths**:

First of all, the article is well written and the main ideas and contributions are easy to follow. The new "DMaaPx" procedure seems very promising to improve the general performances of a VAE. Next, the practical interest of a VAE, in particular the dimensionality reduction providing a compressed representation of high-dimensional complex data, is really well explained and motivated. Moreover, I really appreciated the descriptions of both methods A and B in Section 4. Arguments in favor of using a diffusion model rather than data augmentation are convincing and illustrated numerically in Section 5.

**Weaknesses**:

Since it is the main point of this article, I would first appreciate in the introduction more explanations on why the encoder is more susceptible to overfit than the decoder. It would add even more strenght to the motivation of the article.

Moreover, once the diffusion model is trained, I don't really understand when the new data are generated according to $p_{DM}$ and how they are used during the training process. It would then highligh the contribution to present the "DMaaPx" procedure as a pseudo-code for example.

In addition, my main methodological concern about the new procedure is the training of the diffusion model. Indeed, we can see in Appedix D that the diffusion model is trained on 200,000,000 data. It is a huge amount of data which might not be easily computationally tractable. The resulting diffusion model is then a very good approximation of $p_{data}$ and can explain the very good results of the "DMaaPx" procedure. Thus, it would be very beneficial for the impact of the contribution to evaluate the numerical performances of the "DMaaPx" procedure when training the diffusion model with a more reasonnable amount of data. If the corresponding results are satisfying, the "DMaaPx" procedure may could prevent overfitting even in a small data context.


**Questions and perspectives**:

Here are some additional questions and remarks I was wondering during the reading :
- It is mentionned that a diffusion model is less susceptible to overfit than a VAE. With a more reasonnable amount of training data, can the diffusion model overfit ? And if not, why ?
- Are the metrics in Section 2 proposed in the article or are they extracted from other articles ? If so, some additional references would be appreciated to clarify this point.
- How can we compare the proposed contributions to classical methods to prevent overfitting, like early stopping or dropout for example ? A little discusion on it can be beneficial for the article.
- At last, can the new "DMaaPx" procedure be a remedy to the posterior collapse phenomenon [1,2] which can badly affect the performances of a VAE ? It would be interesting to mention it at least as a perspective.

**References**:

[1] Samuel R Bowman, Luke Vilnis, Oriol Vinyals, Andrew M Dai, Rafal Jozefowicz, and Samy Bengio. Generating sentences from a continuous space. arXiv preprint arXiv:1511.06349, 2015

[2] Junxian He, Daniel Spokoyny, Graham Neubig, and Taylor Berg-Kirkpatrick. Lagging inference networks and posterior collapse in variational autoencoders. arXiv preprint arXiv:1901.05534, 2019

---

> ### Author Response · Authors · 2024-11-29
> **Response from the authors (Part 1)**
>
> > **Q1: More explanations on why the encoder is more susceptible to overfit than the decoder in the introduction.**
>
> **A1:**
> Thanks for the suggestions! We will clarify in the paper that it is an empirical finding that the encoder overfits more than the decoder (Wu et al., 2017;  Cremer et al., 2018). One possible explanation is that since a VAE is normally trained with a finite dataset, due to the ELBO objective and the training algorithm, the encoder is fed with the same data at each epoch. By contrast, the decoder is fed with random samples from $q(z|x)$ and is therefore less likely to see the same data during training. However, this is not saying that the decoder will not be overfitted. Overfitting in a VAE happens in both the encoder and decoder. Our experiments (Fig.2 and Fig.5) show that DMaaPx and overparameterization can alleviate overfitting in both the encoder and the decoder.
>
> In the new version of the paper, we have **updated** the first paragraph in the introduction correspondingly.
>
> ---
>
> > **Q2:  Present the "DMaaPx" procedure as a pseudo-code, so it is clear when the new data are generated according to p_{\text{DM}} and how they are used during the training process.**
>
> **A2:**
> Great suggestion! We have **added** a section (**B.10 Algorithms for Training VAEs with DMaaPx**) in the appendix showing the pseudo-code algorithm for both the normal VAE training and the DMaaPx training procedure for comparison.
>
> ---
>
> > **Q3: “... we can see in Appedix D that the diffusion model is trained on 200,000,000 data.”**
>
> **A3:**
> Thanks for spotting this. Our original text was misleading. What we wanted to say is that we trained the diffusion model with the same number of steps as in (Karras et al., 2022). The models were trained on **50,000 training data** (i.e., the size of the original CIFAR-10 training set) for 4000 epochs. We observed that the training has converged within less than 1000 epochs, but in order to reproduce the same performance as the EDM model in (Karras et al., 2022), we kept the configuration of the number of training epochs unchanged (i.e., 4000 epochs).
>
> We have **updated** the second paragraph of Appendix D correspondingly in the new version of the paper.
>
> ---
>
> > **Q4: it would be very beneficial for the impact of the contribution to evaluate the numerical performances of the "DMaaPx" procedure when training the diffusion model with a more reasonnable amount of data.**
>
> **A4:**
> As we has clarified in **A3**, our diffusion models (DMs) were trained on 50,000 training data. However, the reviewer does raise a very good question. We were also curious how the DMaaPx method will perform if we train the diffusion model with less than 50,000 training data.
>
> Hence, we **added** experiments in the appendix **B.9 Training the Diffusion Model on Subsets of the Training Set**, in which we evaluated the performance of DMaaPx with the diffusion models trained on a {10%, 30%, 50%, 70%, 90%} subset of the original CIFAR-10 training data (of size 50,000).
>
> Figure 16 shows the resulting generalization gaps. The horizontal dashed and solid blue lines correspond to the train and test performance of the VAE using normal training (i.e., trained with the full CIFAR-10 and without DM). We can see that when using the 30% subset to train DM, DMaaPx already matches the test performance of normal training. In other words, for DMaaPx, using only a subset of 15,000 training samples matches the performance of normal training with the full training set of size 50,000. At the same time DMaaPx has a significantly smaller generalization gap.
>
> Another observation is that using a larger subset for training the DM keeps improving the performance of DMaaPx until reaching diminishing returns for $\geq$ 70% of CIFAR-10.
>
> ---
>
> > **Q5: With a more reasonnable amount of training data, can the diffusion model overfit? And if not, why?**
>
> **A5:**
> Our paper is not trying to discuss overfitting in diffusion models. Nevertheless, diffusion models (DMs) can be overfitted. As for how an overfitted diffusion model might impact the DMaaPx method, we can consider the following two factors:
>
> In the extreme case of overfitting for a DM, it will memorize all the training data exactly, which means DMaaPx will degenerate back to normal training, hence it will not perform worse than the normal training baseline.
>
> In addition, as we discussed in the answer of last question (i.e., **A4**), we empirically show that even if we reduce the training data for a DM significantly from 50,000 to 15,000, DMaaPx still matches the performance of normal training with 50,000 data.

---

> ### Author Response · Authors · 2024-11-29
> **Response from the authors (Part 2)**
>
> > **Q6: Are the metrics in Section 2 proposed in the article or are they extracted from other articles?**
>
> **A6:**
> We do not treat these metrics as our contributions.
> Our definition of the generalization gap is based on the well accepted definition of generalization in machine learning, which is the difference between the training loss and the test loss.
> Our definition of the amortization gap is the same as in (Cremer et al., 2018), which is among the first investigations of overfitting in VAEs. We **updated** the paper by adding the citation right before Equation (4).
> Our definition of the robustness gap is based on (Kuzina et al., 2022), which has been cited before Equation (5).
>
> ---
>
> > **Q7: How can we compare the proposed contributions to classical methods to prevent overfitting, like early stopping or dropout for example?**
>
> **A7:**
> Our paper did already have comparisons to a few classical methods: (1) **Data augmentation** is used as a baseline in most of our experiments, which is classically the most straightforward method to prevent overfitting in image related models. (2) **Early stopping** can be inferred from the existing plots as the x-axis in Figure 2 and Figure 5 are training epochs for the normal training baseline. For example, in Figure 2, we can see DMaaPx still outperforms the normal training baseline in all three gaps if we use the “forking point” (i.e., around 200 epochs) for early stopping, which is when the test performance starts decreasing for normal training. (3) **Prior work** on preventing overfitting specifically for VAEs is also compared in Section 5.2 Figure 4. And we showed that DMaaPx is better. (4) **Dropout** is not compared, because it will introduce additional random variables in the model (VAEs normally only assume two random variables $z$ and $x$), which will result in a different modeling assumption than the normal training baseline. Note that due to the same reason, dropout is commonly avoided in VAEs when they are used for modeling some underlying data distributions.
>
> ---
>
> > **Q8: Can the new "DMaaPx" procedure be a remedy to the posterior collapse phenomenon which can badly affect the performance of a VAE?**
>
> **A8:**
> Great question! As far as we know, the relationship between posterior collapse and the size of training data are not well explored for VAEs at the moment. What we do know is that posterior collapse is commonly believed to be related to the strength of the KL term in the ELBO (i.e., the $\beta$ parameter in $\beta$-VAE), and an extremely large $\beta$ will result in posterior collapse such that the posterior is forced to be the same as the prior. This can lead to a bad generalization performance, no matter how large the training is. Therefore, DMaaPx might not be the solution to the posterior collapse phenomenon. We believe posterior collapse should be studied from a different direction with different tools.
>
> ---
>
> > **Q9: I would appreciate to change $p(z)$ to $p_{Z}(z)$ in order to clearly distinguish the prior distribution to the one of the data.**
>
> **A9:**
> We have **updated** the notation as requested in the new version of the paper.

---

### Review · Reviewer_NGVK · 2024-11-19

**Summary Of Contributions:**

This paper explores improving VAE generalization using synthetic data from diffusion models and overparameterization. It shows synthetic data reduces overfitting better than augmentation, and increasing latent parameters enhances performance, revealing practical insights and the double descent phenomenon.

**Audience:**

Yes

**Claims And Evidence:**

Yes

**Requested Changes:**

- Typo: "double-decent" should be "double-descent" in the last paragraph.

**Strengths And Weaknesses:**

Strengths:
- The paper is well organized and clearly presented.
- The paper offers practical guidelines for improving VAEs, such as prioritizing latent dimension parameterization and effectively using synthetic data.
- The paper provides valuable insights into the double descent phenomenon for VAEs.


Weaknesses:
- Experiments are limited to image datasets, limiting generalization to other domains.
- DMaaPx implies that diffusion models are better than other types of generative models (e.g., VAE, GANs) at capturing the real data distribution. Additional theoretical or experimental evidence would strengthen the paper.

Questions:
- The paper mentions training diffusion models on CIFAR-10/BinaryMNIST/FashionMNIST for 2.5 days on 8 GPUs. Given the small sizes of these datasets, is this a mistake? If accurate, the computational cost is orders of magnitude higher than training the VAE itself. Is this significant overhead worthwhile? Is it computationally efficient in terms of learning representation compared to other methods?

---

> ### Author Response · Authors · 2024-11-29
> **Response from the authors**
>
> > **Q1: Experiments are limited to image datasets, limiting generalization to other domains.**
>
> **A1:**
> Indeed, our experiments are only in the image domain. The investigated DMaaPx relies on the quality of the pre-trained diffusion model. While there currently is a lot of research into diffusion models in domains other than images, the image domain has (so far) received most of the research attention. Therefore, we expect diffusion models trained on images to show the best generation performance. As a result, we intentionally set the scope of this paper to the image domain to leverage the research in image diffusion models and  to have a restricted domain that we can investigate extensively. However, we think that DMaaPx or over-parameterization can have a significant impact on performance in domains other than images. We will leave an investigation of other data types as an interesting direction for future research.
>
> ---
>
> > **Q2: DMaaPx implies that diffusion models are better than other types of generative models (e.g., VAE, GANs) at capturing the real data distribution. Additional theoretical or experimental evidence would strengthen the paper.**
>
> **A2:**
> There is not a single metric that’s absolutely “best” for measuring the goodness of a generative model (Bischoff et al., 2024). In the image domain, it is well accepted that Fréchet Inception Distance (FID) and Inception Score (IS) are the go to metrics for benchmarking a generative model. Under these two metrics, diffusion models (DMs) have shown to be the best at the moment with a large amount of empirical evidence (please see the survey paper (Yang et al., 2023) for broader evidence).
>
> The goal of our paper does not contain showing whether the DM we used in the paper is better at approximating $p_{data}(x)$ then the other generative models. We want to highlight that the key takeaway here is that if we have a generative model that’s good at capturing $p_{data}(x)$, the synthetic data from this model can help improve the generalization of VAEs, without using other tricks from prior works. And, the current state-of-the-art DM is already showing the usefulness of DMaaPx. It is very likely there will be better generative models in the future, which are more accurate in approximating $p_{data}(x)$ by some metrics. By then, we can simply replace the DM with such new generative models, and the method DMaaPx will still work due to the arguments in Section 4.1.
>
> ---
>
> Bischoff et al., “A Practical Guide to Sample-based Statistical Distances for Evaluating Generative Models in Science”, 2024
>
> Yang et al., “Diffusion Models: A Comprehensive Survey of Methods and Applications”, 2023
>
> ---
>
> > **Q3: The paper mentions training diffusion models on CIFAR-10/BinaryMNIST/FashionMNIST for 2.5 days on 8 GPUs. Given the small sizes of these datasets, is this a mistake? If accurate, the computational cost is orders of magnitude higher than training the VAE itself. Is this significant overhead worthwhile? Is it computationally efficient in terms of learning representation compared to other methods?**
>
> **A3:**
> There is indeed a mistake, the diffusion model (DM) we used, i.e., EDM (Karras et al., 2022), was trained for 4000 epochs in the original paper (Karras et al., 2022), which is about 1 day on 8 GPUs. We observed that the training has converged within less than 1000 epochs, but in order to reproduce the same performance as the original paper (Karras et al., 2022), we keep the configuration of the number of training epochs unchanged (i.e., 4000 epochs). We have **updated** the second paragraph of Appendix D correspondingly in the new version of the paper.
>
> Nevertheless, training a DM from scratch is not cheap. But there is a growing amount of pre-trained diffusion models publicly available (e.g., on Huggingface) that can be utilized for sampling (without investing any computation for training). If one uses a pre-trained diffusion model, the overhead reduces to sampling the diffusion model. In practice, before training your own DMs, one should always search for existing publicly available DMs online (e.g., HuggingFace). If we use pretrained DMs and require only small amount of samples ($k=5$ in Figure 3 right, fully parallelizable), this process is very much feasiable with negligible overhead.
>
> Additionally, DMs and their sampling processes currently get a significant amount of attention with the goal to improve the sampling time (e.g., consistency models, step-aware models, various improvements to the U-Net, other dynamics, …).
>
> Overall, our paper investigates the potential benefits of using synthetic data and overparameterization for VAEs training if we have enough compute budget, but whether these methods are worthwhile really depends on the goal and the compute budget of the particular practitioner.
>
> ---
>
> > **Q4: Typo: "double-decent" should be "double-descent" in the last paragraph.**
>
> **A4:**
> Thanks! We fixed the typo in the **updated** version of the paper.

---

### Author Response · Authors · 2024-11-29
**General Response: Additional Experiments**

We would like to thank all reviewers for taking the time to review our work.
In response to reviewer feedback we included **4** additional experiments in the revised version of our paper.

---

1. **Smoothness of Latent Space (Appendix  B.6)**

In response to a question by the reviewers, we added an experiment comparing the smoothness of the latent space between DMaaPx and normal training: We linearly interpolate between two latents mapped from two data samples and evaluate the corresponding densities. The results show that the densities of the interpolations in the DMaaPx trained latent space are generally higher and indeed much smoother than for the normally trained VAE.

2. **Mixing Synthetic Data with Real Data (Appendix  B.8)**

Upon request by the reviewers, we added an ablation study evaluating the performance of VAEs trained with $n \in$ {0%, 20%, 40%, 60%, 80%, 100%} of real data and $100-n%$ of synthetic data. We ran these experiments with and without augmentation (i.e., Aug.Tuned). Overall, we don’t see too much benefit of mixing real data with synthetic data if we can generate as many synthetic samples as needed. Similarly, adding augmentation on top of synthetic data does not bring additional gain.


3. **Training Diffusion Model on Subsets of the Training Set (Appendix B.9)**

Inspired by a comment from the reviewers, we added an evaluation of the performance of DMaaPx using a diffusion model that has been trained on a {10%, 30%, 50%, 70%, 90%} subset of the original CIFAR-10 training data. We find that already 30% of the training set matches the test performance of normal training. At the same time DMaaPx has a significantly smaller generalization gap.

4. **Discriminating Synthetic and Real Samples by Classification (Appendix  D.3)**

Motivated by a question from the reviewers, we added experiments of a classifier distinguishing between real and synthetic data samples. We compare samples from five diffusion models trained on {30%, 50%, 70%, 90%, 100%} of the original CIFAR-10 training data. We can see that diffusion models trained with more data produce samples that are harder to distinguish (the test accuracy for discriminating between synthetic and real data drops). Using 100% training data corresponds to around 56% test accuracy. We conclude that the diffusion model might not be perfect but is chosen as the current state of the art that has been well recognized by the community (Karras et al., 2022).

---

We think the results in **Smoothness of Latent Space (B.6)** are quite interesting, and will move this section to the main text later.

---

### Decision · Action_Editor_igg8 · 2024-12-19

**Recommendation:** Accept as is

**Comment:**

The authors have addressed the comments of the reviewers in the revision. I do not request additional changes and suggest to accept the paper as is.

**Audience:**

I agree with the reviewers that the submission provides interesting insights, and therefore it can be safely concluded that at least some individuals of the TMLR's audience would be interested in the findings.

**Claims And Evidence:**

The claimed benefits in terms of generalization and robustness when training a VAE with synthetic samples from a learned (diffusion) model compared to naive data augmentation  are both interesting and supported by evidence. The submission makes additional claims regarding the different effects on generalization when changing the number of parameters (independently on the encoder or decoder) and suggest a double descent behavior, which are also sufficiently well supported.